# Serial femtosecond crystallography on in vivo-grown crystals drives elucidation of mosquitocidal Cyt1Aa bioactivation cascade

Guillaume Tetreau [1,13], Anne-Sophie Banneville[1,13], Elena A. Andreeva [1,13], Aaron S. Brewster [2,13], Mark S. Hunter[3], Raymond G. Sierra [3], Jean-Marie Teulon[1], Iris D. Young [2], Niamh Burke[1], Tilman A. Grünewald [4], Joël Beaudouin[1], Irina Snigireva [4], Maria Teresa Fernandez-Luna[5,12], Alister Burt [1], Hyun-Woo Park[5,6], Luca Signor [1], Jayesh A. Bafna[7], Rabia Sadir[1], Daphna Fenel[1], Elisabetta Boeri-Erba[1], Maria Bacia[1], Ninon Zala[1], Frédéric Laporte [8], Laurence Després [8], Martin Weik [1], Sébastien Boutet [3], Martin Rosenthal [4], Nicolas Coquelle [9], Manfred Burghammer[4], Duilio Cascio[10], Michael R. Sawaya [10], Mathias Winterhalter [8], Enrico Gratton[11], Irina Gutsche [1], Brian Federici[5], Jean-Luc Pellequer [1], Nicholas K. Sauter[2] & Jacques-Philippe Colletier [1✉]

Cyt1Aa is the one of four crystalline protoxins produced by mosquitocidal bacterium *Bacillus thuringiensis israelensis* (*Bti*) that has been shown to delay the evolution of insect resistance in the field. Limiting our understanding of *Bti* efficacy and the path to improved toxicity and spectrum has been ignorance of how Cyt1Aa crystallizes in vivo and of its mechanism of toxicity. Here, we use serial femtosecond crystallography to determine the Cyt1Aa protoxin structure from sub-micron-sized crystals produced in *Bti*. Structures determined under various pH/redox conditions illuminate the role played by previously uncharacterized disulfide-bridge and domain-swapped interfaces from crystal formation in *Bti* to dissolution in the larval mosquito midgut. Biochemical, toxicological and biophysical methods enable the deconvolution of key steps in the Cyt1Aa bioactivation cascade. We additionally show that the size, shape, production yield, pH sensitivity and toxicity of Cyt1Aa crystals grown in *Bti* can be controlled by single atom substitution.

[1] Univ. Grenoble Alpes, CNRS, CEA, Institut de Biologie Structurale, Grenoble F-38000, France. [2] Molecular Biophysics and Integrated Bioimaging Division, Lawrence Berkeley National Laboratory, Berkeley, CA 94720, USA. [3] Linac Coherent Light Source, SLAC National Accelerator Laboratory, Menlo Park, CA 94025, USA. [4] European Synchrotron Radiation Facility (ESRF), BP 220, Grenoble 38043, France. [5] Department of Entomology and Institute for Integrative Genome Biology, University of California, Riverside, CA 92521, USA. [6] Department of Biological Sciences, California Baptist University, Riverside, CA 92504, USA. [7] Department of Life Sciences & Chemistry, Jacobs University, Bremen, Germany. [8] Univ. Grenoble Alpes, CNRS, LECA, Grenoble F-38000, France. [9] Large-Scale Structures Group, Institut Laue-Langevin, Grenoble F- 38000, France. [10] UCLA-DOE Institute for Genomics and Proteomics, Department of Biological Chemistry, University of California, Los Angeles, CA 90095-1570, USA. [11] The Henry Samueli School of Engineering, University of California, Irvine Irvine, CA 92697-2715, USA. [12] Present address: Department of Biology, Baylor University, One Bear place 97388, Waco, TX 76798-7388, USA. [13] These authors contributed equally: Guillaume Tetreau, Anne-Sophie Banneville, Elena A. Andreeva, Aaron S. Brewster ✉email: colletier@ibs.fr

Mosquitoes remain the organisms most harmful to human health, transmitting diseases such as malaria, dengue fever, and filariasis. Disease prevention relies mostly on the control of mosquito vector populations by use of chemical insecticides but these elicit resistance whilst also harming crustaceans, bees, and fish. A safer alternative to chemicals is the dissemination of sporulated formulations of the mosquitocidal bacterium, *Bacillus thuringiensis* subspecies *israelensis* (*Bti*), which upon ingestion by mosquito larvae destroys their midgut, killing them[1]. The active ingredient of *Bti* is a parasporal body that contains three natural sub-micron-sized crystals of four highly efficient mosquito-specific toxins, namely Cyt1Aa, Cry11Aa and co-crystallizing Cry4Aa and Cry4Ba. Following ingestion by mosquito larvae, the crystals promptly dissolve in the alkaline environment of the gut (pH ~11), releasing protoxins that are activated into toxins through proteolytic cleavage of propeptides by gut enzymes. Whereas Cyt1Aa directly interacts with lipids from gut cell membranes, Cry toxins require interaction with membrane-bound receptors. Each activated protein eventually self-assembles into cytolytic oligomers that perforate gut cells, leading to mosquito larvae death[1]. Cyt1Aa oligomers can additionally serve as substitution receptors for Cry toxins, enabling them to kill cells even in the absence of mosquito receptors. This synergy explains how *Bti* is able to evade resistance and makes Cyt1Aa the key element of its mosquitocidal arsenal. However, the molecular determinants of Cyt1Aa crystallization in *Bti* cells and of crystal dissolution in the mosquito midgut remain unclear, and the mechanism by which oligomers form and exert direct toxicity to mosquito gut cells is actively debated[2,3]. Most investigators agree that Cyt1Aa binds directly to lipids and acts as a receptor for *Bti* Cry toxins. However, two contrasting models—not necessarily incompatible—have been proposed for toxicity[2,3]. In one, Cyt1Aa monomers assemble into monovalent cation-selective channels with a radius of 6–20 Å spanning through the membrane, thereby killing cells by ion leakage ("pore-forming" model)[4–6]. In the other, monomers form oligomers on the membrane surface causing lipid faults that result in cell lysis ("detergent" model)[7].

Here, we use serial femtosecond crystallography (SFX) at an X-ray free electron laser (XFEL) to identify the features governing the in vivo crystallization of Cyt1Aa in *Bti* cells, and to track the structural dynamics driving natural crystal dissolution in the alkaline mosquito larvae gut. Use of complementary biophysical, biochemical, and toxicological methods in combination with mutagenesis at positions identified in difference electron density maps allows us to formulate a new model for the Cyt1Aa bioactivation cascade, from self-inhibition and in vivo crystallization in *Bti* cells to oligomer formation and cell lysis. By illustrating how the crystal properties and toxicity of naturally crystalline Cyt1Aa can be tuned by single atom substitutions at strategic positions, our results pave the way for a rational tailoring of its properties to human needs.

## Results

### The N-terminal propeptide governs in vivo crystallization.

We produced sub-micron-sized crystals of Cyt1Aa in vivo by recombinant expression in an acrystalliferous strain of *Bti* (4Q7)[8]. Crystal size, shape, integrity, and diffraction power were assessed by scanning electron microscopy (SEM) (Fig. 1a), atomic force microscopy (AFM) (Fig. 1b), transmission electron microscopy (TEM) (Fig. 1c), and serial synchrotron crystallography, respectively, revealing highly ordered bipyramidal crystals ($0.6 \times 0.6 \times 0.9\ \mu m^3$, corresponding to ~180,000 unit cells) which diffract to ~4.0 Å resolution when exposed to a sub-microfocus synchrotron X-ray beam (0.7 μm FWHM; ESRF-ID13) at 100 K[9]

(Supplementary Fig. 1). Using SFX at the CXI-SC3 micro-focused beamline[10] of the Stanford Linear Accelerator Center (SLAC) Linac Coherent Light Source (LCLS), diffraction data extending to 1.86 Å could be collected at room temperature (RT) for the wild-type (WT) Cyt1Aa protoxin at pH 7 ("pH7" dataset, Table 1), from sub-micron-sized crystals injected by a MESH device[11]. Data were phased by molecular replacement using as a starting model the in vitro structure of proteolytically activated Cyt1Aa[12] (PDB entry: "3ron [10.2210/pdb3RON/pdb]") determined by synchrotron crystallography at 100 K. Therefore, 37 residues from the N-terminus and 7 residues from the C-terminus were missing from the initial model, respectively, corresponding to the digested propeptides. Clear residual density allowed us to build most of these residues (H6-L249) (Fig. 1d). At the monomeric level, the Cyt1Aa protoxin displays a conformation overall similar to that of the activated toxin, with two outer layers of α-helix hairpins, αA/αB and αC/αD, respectively covering the hydrophilic and hydrophobic sides of a central five-stranded mixed β-sheet (namely β2-β5-β6-β7-β3; Fig. 1e). Besides the presence of propeptides, the largest structural differences between the protoxin and the toxin are observed at the C-terminus where αF residues display distinct conformations (Fig. 1e). Additionally, the αC/αD hairpin draws away from the β-sheet and from helix αE upon activation (Fig. 1e and Supplementary Fig. 2a, b). While these conformational changes are on the overall subtle, they appear to be concerted at the main-chain level (Supplementary Fig. 2b). At the side chain level, all residues assuming different conformations are found on the αC/αD side of the β-sheet, except D72 (Fig. 1e). In some cases, we observe conformational changes that maintain an H-bond in place, e.g. that which tethers the αC/αD hairpin (Q138) to the tip of β2 (E45) (Supplementary Fig. 2c).

In vivo-grown Cyt1Aa protoxin crystals are remarkably packed, burying 40.1% of surface area at crystal contacts ($5484.2\ Å^2$ out of $13661.2\ Å^2$ of total monomeric areas), compared with only 13.3% ($1252.7\ Å^2$ out of $9405.5\ Å^2$) in crystals of the activated toxin grown in vitro. Examination at the unit-cell level reveals that the Cyt1Aa protoxin crystallizes as a domain-swapped (DS) dimer[13] with strands β1 (contributed by the N-terminal propeptide) and β2 entwined at the DS interface (interface #1), aligned with a crystallographic two-fold axis (Fig. 2a, Supplementary Fig. 3 and Supplementary Table 1). The DS interface of Cyt1Aa, which is mainly stabilized by H-bonds between carboxylic (E32, E45, D137, E156), amide (Q138, N181) and amine (R30) groups (Fig. 2b), also includes the αC/αD hairpin and the C-terminal propeptide helix αF (Fig. 2a), burying a cumulated surface area of $3077\ Å^2$ in each monomer—i.e. 22.5 % of the protoxin accessible area, which is more than all other crystal packing interfaces combined ($2414\ Å^2$ over nine interfaces; Supplementary Fig. 3 and Supplementary Table 1) or all interfaces in the in vitro crystals of the activated toxin combined ($1710 \pm 120\ Å^2$) (Fig. 1f). Formation of the DS dimer thus likely precedes crystallization in *Bti* cells. A similar DS interface was previously observed in the in vitro structure of the homologous non-cytolytic Cyt2Ba protoxin (33% identity with Cyt1Aa), but its biological significance could not be ascertained[14].

Strikingly, residues absent from the activated toxin structure are the main contributors to the packing of the natural crystals. Most notably, the N-terminal propeptide buries 73% of its surface area ($2704$ over $3691\ Å^2$) across nine interaction zones, contributing 34 H-bonds, 10 salt-bridges and a disulfide bridge at position C7 (Fig. 1g). This disulfide bridge is aligned with the other crystallographic two-fold axis (Fig. 2). Collection of a dataset upon soaking of crystals with 1 mM DTT ("DTT" dataset) indicates that at pH 7, rupture of the C7(Sγ)-C7(Sγ), evidenced by a strong negative peak in the $Fo^{DTT,\ pH7}-Fo^{pH7}$ map, is not

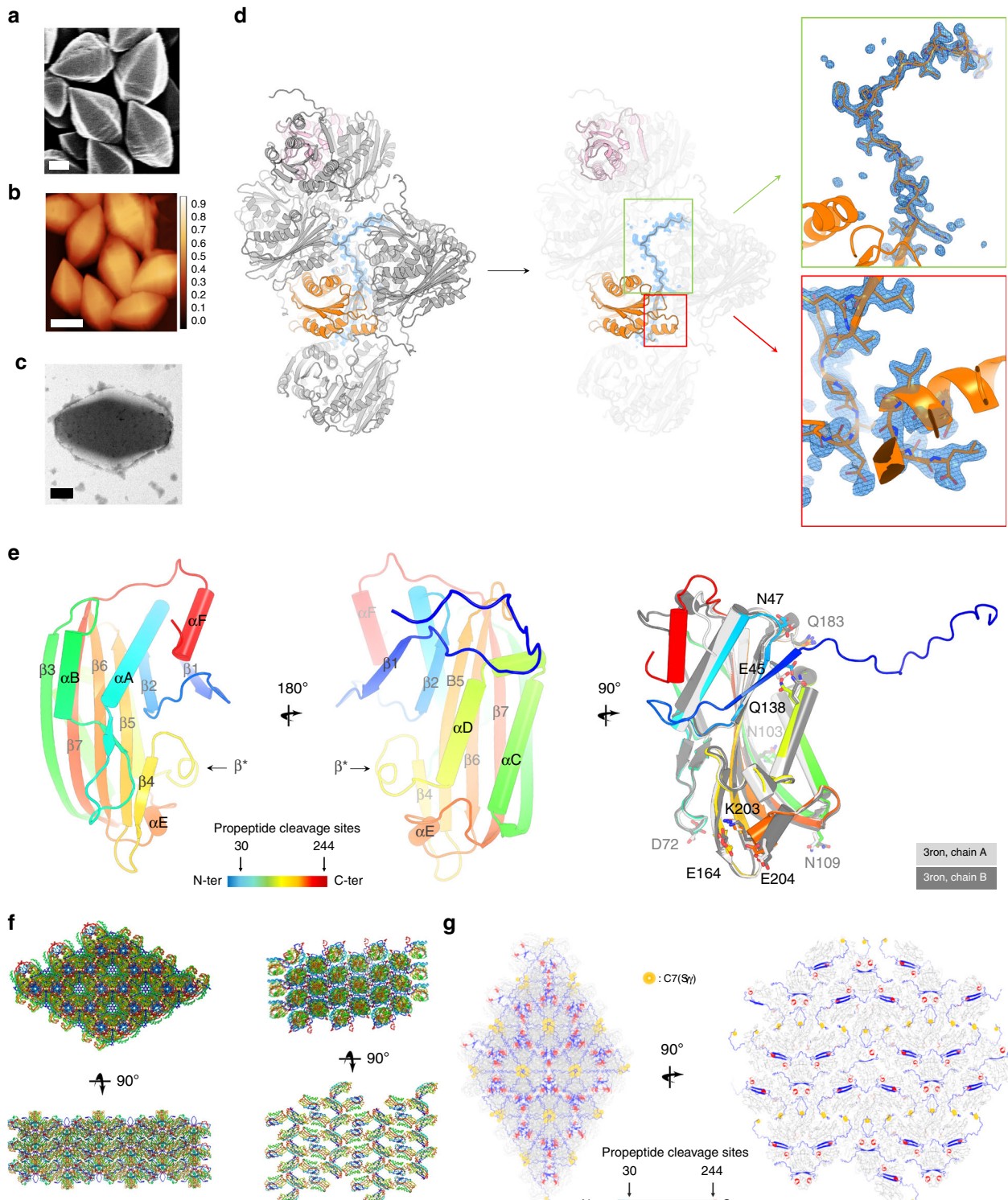

**Fig. 1 SFX on in vivo-grown crystals enables determination of the Cyt1Aa protoxin structure. a–c** Cyt1Aa sub-micron-sized crystals were grown in vivo by recombinant expression in *Bti*-4Q7, and their quality assessed by scanning electron microscopy (SEM; **a** scale bar = 200 nm), atomic force microscopy (AFM; **b** scale bar = 500 nm) and transmission electron microscopy (TEM; **c** scale bar = 200 nm). **d** Electron density was visible for the propeptides in the initial 2Fo-Fc electron density map displayed at 1 σ. The N-terminal propeptide establishes contact with four symmetry-related molecules. **e** Tertiary structure of Cyt1Aa. In the right panel, the two chains constitutive of the asymmetric unit of the activated toxin ("3ron [10.2210/pdb3RON/pdb]") are overlaid on the protoxin structure, with residues displaying different side chain conformations shown as sticks. Secondary structure information is overlaid on the models. **f** Packing in the natural protoxin crystals grown in vivo (left) and in the crystals of the activated toxin grown in vitro (right). **g** The N-terminal and C-terminal propeptides scaffold the natural crystals.

**Table 1 Data collection and refinement statistics (molecular replacement).**

|  | Wild type, pH 7 | Wild type, DTT | Wild type, pH 10 | C7S, pH 7 |
|---|---|---|---|---|
| PDB accession code | 6T14 | 6T19 | 6T1A | 6T1C |
| *Data collection* |  |  |  |  |
| Space group | $P6_122$ | $P6_122$ | $P6_122$ | $P6_122$ |
| Cell dimensions |  |  |  |  |
| $a, b, c$ (Å) | 64.8, 64.8, 164.5 | 65.6, 65.6, 164.3 | 65.5, 65.5, 165.5 | 65.6, 65.6, 164.1 |
| $\alpha, \beta, \gamma$ (°) | 90, 90, 120 | 90, 90, 120 | 90, 90, 120 | 90, 90, 120 |
| X-ray beam focus (μm) | 1.3 | 1.3 | 1.3 | 1.3 |
| Number of collected frames | 65473 | 39547 | 45117 | 18218 |
| Number of indexed patterns | 8584 | 18882 | 20052 | 7754 |
| Number of merged images | 8462 [b] | 18766 [b] | 19924 [b] | 7683 [b] |
| Resolution (Å)[a] | 43.5-1.86 (1.89-1.86) | 43.5-1.83 (1.86-1.83) | 43.5-1.85 (1.88-1.85) | 43.5-1.97 (2.01-1.97) |
| Number of observations | 2003832 (16132) | 3285436 (30913) | 3123781 (30982) | 1203324 (23089) |
| Number of unique reflections | 17966 (898) | 18261 (1003) | 18439 (910) | 14420 (877) |
| $I/\sigma(I)$[a] | 18.6 (1.6) | 32.8 (3.0) | 34.6 (3.2) | 19.8 (2.8) |
| Rsplit (%)[a] | 12.5 (87.7) | 7.4 (51.7) | 7.0 (49.5) | 12.5 (61.6) |
| $CC_{1/2}$[a] | 99.5 (7.8) | 99.8 (16.4) | 99.8 (7.3) | 99.4 (17.8) |
| Completeness (%)[a] | 98.7 (100.0) | 99.9 (100.0) | 99.9 (100.0) | 99.9 (100.0) |
| Multiplicity[a] | 107.8 (18.0) | 160.5 (30.8) | 165.4 (33.6) | 65.8 (26.33) |
| *Refinement* |  |  |  |  |
| Resolution (Å)[a] | 1.86 (1.91-1.86) | 1.85 (1.90-1.85) | 1.85 (1.90-1.85) | 2.00 (2.05-2.00) |
| Number of reflections[a] | 16099 (1126) | 17339 (1231) | 17513 (1262) | 13698 (722) |
| $R_{work}$ /$R_{free}$[a] | 0.21.7 (0.503)/0.25.9 (0.451) | 0.231 (0.459)/0.295 (0.547) | 0.237 (0.497)/0.287 (0.494) | 0.220 (0.427)/0.267 (0.473) |
| Number of atoms |  |  |  |  |
| Protein | 1923 | 1941 | 1987 | 1941 |
| Water | 251 | 293 | 281 | 276 |
| B-factors (Å²) |  |  |  |  |
| Protein | 30.2 | 32.6 | 36.1 | 35.8 |
| Water | 47.7 | 52.3 | 48.5 | 48.0 |
| R.m.s. deviations |  |  |  |  |
| Bond lengths (Å) | 0.007 | 0.007 | 0.007 | 0.007 |
| Bond angles (°) | 1.146 | 1.083 | 1.103 | 1.120 |

[a]Values in parentheses are for highest-resolution shell.
[b]The number of merged images corresponds to number of crystals used to solve the structure.

sufficient to prompt crystal dissolution, degrade crystal quality or induce structural changes within the protoxin (Fig. 2b, Supplementary Fig. 4 and Table 1). Collection of a dataset upon soaking of crystals in CAPS buffer at pH 10.3 ("pH10" dataset) identifies the DS interface as the most sensitive to pH elevation, with the strongest positive and negative peaks in the $Fo^{pH10}$–$Fo^{pH7}$ map observed on E32, E45, and Q138 (Fig. 2b and Supplementary Fig. 4). Refinement of the "pH10" structure confirms changes in interactions at the DS interface (Fig. 2b and Supplementary Fig. 2a–e and Supplementary Fig. 4), with loss of H-bonds between facing monomers (e.g. between E32 and E45 side chains), as well as within each monomer (e.g. between E45 and Q138 side chains). As a result, strands β1, β2 and the C-terminal propeptide concertedly draw away from strand β3, at the opposite end of the β-sheet, but also from the αC, αD, and αE helices, which together cover the hydrophobic face of the β-sheet (Supplementary Fig. 2a–e). The C7 side chain displays two alternate conformations in the "pH10" structure, respectively characterized by Sγ–Sγ distances of 2.1 and 3.9 Å, indicating partial rupture of the disulfide bridge upon elevation of pH to 10.3 (Fig. 2b and Supplementary Fig. 4). Importantly, a positive peak is seen between D11 and Q168 in both the $Fo^{pH10}$–$Fo^{pH7}$ and the $Fo^{DTT,pH7}$–$Fo^{pH7}$ maps, highlighting that this interface is sensitive to both reducing agent and pH elevation (Fig. 2b and Supplementary Fig. 4). The αA/αB face of the β-sheet is not affected by pH elevation.

We engineered a C7S mutation to probe the role of the disulfide bridge. By recombinant expression in *Bti*-4Q7, we obtained sub-micron-sized crystals diffracting to 2.0 Å resolution,

which revealed a structure nearly indiscernible from the "DTT" structure (Fig. 3 and Supplementary Fig. 2a–e). Thus, neither crystal formation nor crystalline order depends on the disulfide-bridge chaining of DS dimers. This step therefore likely occurs after the crystal is fully formed. We also mutated the other residues pinpointed by strong peaks in the Fo–Fo maps, D11, E32, E45, Q168, and Y171, hypothesizing that they would be central to crystal formation and dissolution, and possibly function (Fig. 3). As a control for C7, we mutated the other cysteine of Cyt1Aa, C190. For all mutants, sub-micron-sized crystals were obtained by recombinant expression in *Bti*-4Q7. A full description of our difference-density based mutation strategy is presented in Supplementary Note 1.

**Mapping the role of the disulfide bridge and DS interface**. The Cyt1Aa bioactivation cascade involves dissolution of crystals in the alkaline environment of the mosquito gut. Accordingly, a pH of $11.2 \pm 1.0$ is required to solubilize 50% of Cyt1Aa WT crystals after 1h incubation ($SP_{50}$) (Fig. 3 and 4a and Supplementary Table 2). This incubation time was chosen on the basis that the transit time along the mosquito larvae gut is 30–60 min, depending on species[15]. When DTT is added, crystals solubilize at a significantly lower pH, with a $SP_{50}$ of $9.8 \pm 1.0$ (Fig. 4a and Supplementary Table 2). The solubilization pattern of C190V crystals does not significantly differ from the WT, confirming that this mutation does not affect the pH sensitivity of crystals and that the effect of DTT on crystal solubilization is unrelated to C190 (Figs. 3 and 4a). Likewise, the hydrophobic-core Y171F

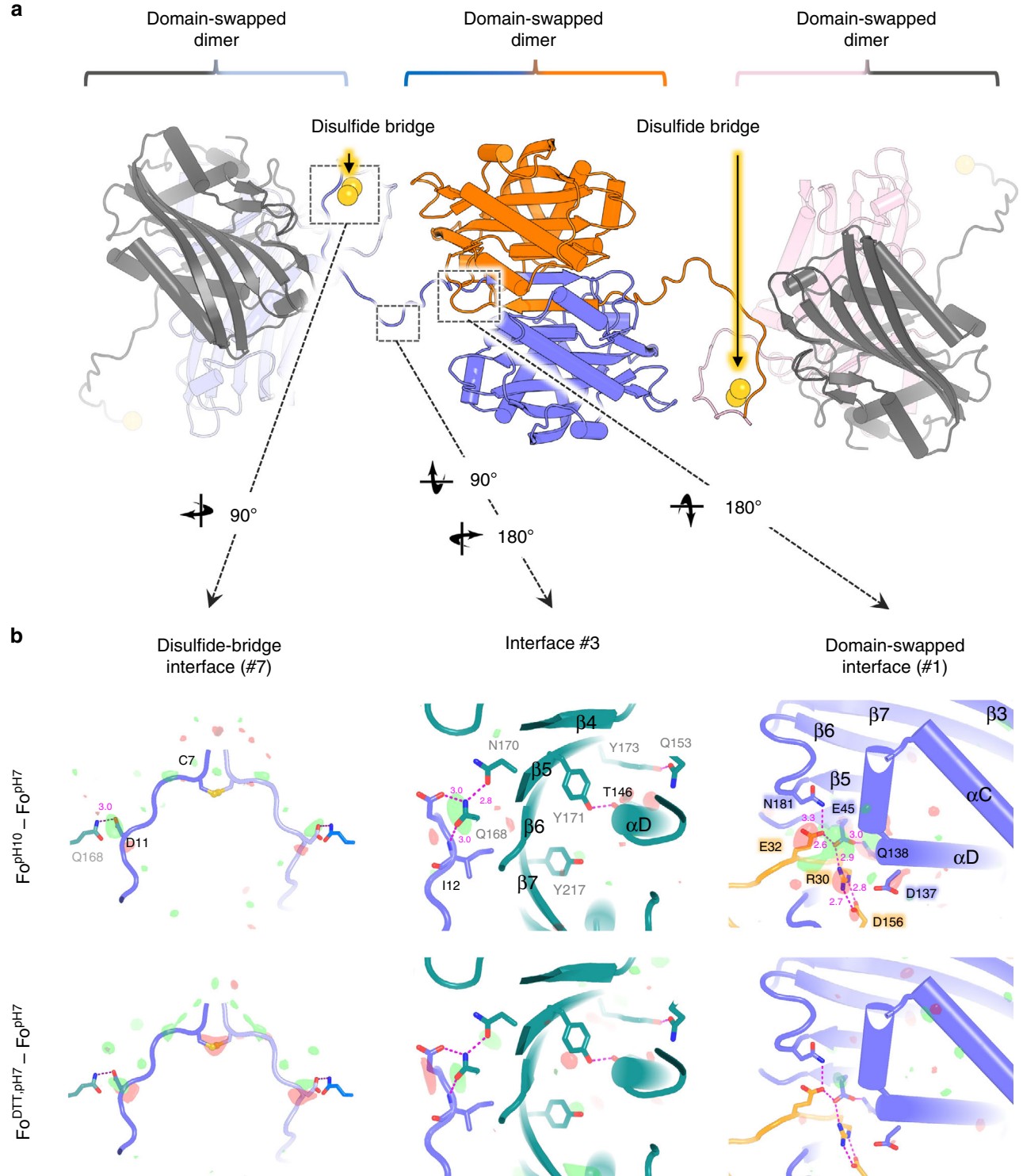

**Fig. 2 Disulfide-bridge chaining of domain-swapped (DS) dimers in the natural Cyt1Aa crystals. a** Cyt1Aa dimers associated through a DS interface constitute the building block of natural crystals. These DS dimers are chained one to another by a disulfide bridge at position C7. **b** Fourier difference maps computed between datasets, and phased by the pH7 structure, highlight the most striking conformational changes upon pH elevation (upper panels; $Fo^{[pH10]}–Fo^{[pH7]}$ map) and DTT soak (lower panels; $Fo^{[DTT]}–Fo^{[pH7]}$ map). The difference maps are overlaid on the pH7 protoxin structure, shown as a slate-colored ribbon. Symmetry related molecules are all colored differently, with each molecule having the same color coding in all panels. From left to right, the figure shows the maps contoured at ±3 σ around the disulfide bridge, at crystal packing interface #3 and at the DS interface, respectively. Positive and negative peaks are shown in green and red. Secondary structure information is overlaid on the models.

| Cyt1Aa mutant | | Wild-type | C7S | C190V | Y171F | D11N | Q168E | E32Q | E45Q |
|---|---|---|---|---|---|---|---|---|---|
| Crystal shape (SEM) | |  |  |  |  |  |  |  |  |
| Crystal size (in nm) (mean ± SE) | $L^1$ | 925.0 ± 28.3a | 746.6 ± 16.8b | 666.8 ± 17.5b | 751.6 ± 14.6b | 1217.2 ± 36.4c | 1086.7 ± 26.6d | 894.9 ± 20.7a | 1153.4 ± 20.8cd |
| | $W^2$ | 591.2 ± 17.3a | 612.4 ± 9.4a | 402.5 ± 8.7b | 482.9 ± 8.5c | 588.2 ± 18.6a | 601.1 ± 13.2a | 619.5 ± 13.8a | 768.0 ± 15.6d |
| | L/W | 1.57 ± 0.15ac | 1.22 ± 0.13b | 1.66 ± 0.13c | 1.56 ± 0.14ac | 2.09 ± 0.27d | 1.81 ± 0.16e | 1.44 ± 0.14a | 1.51 ± 0.12a |
| Productivity[3] | | ++ | ++ | ++ | + | ++ | + | +++ | ++ |
| Solubility at 1 h ($SP_{50}$ ± SE)[4] | $-DTT^5$ | 11.18 ± 1.01ab | 9.58 ± 1.02c | 11.07 ± 1.01ab | 10.90 ± 1.02ab | 10.87 ± 1.01a | 9.95 ± 1.01c | 11.31 ± 1.02ab | 11.38 ± 1.01b |
| | $+DTT^6$ | 9.78 ± 1.01ab | 9.63 ± 1.02ab | 9.71 ± 1.01ab | 9.50 ± 1.02ab | 9.22 ± 1.02b | 8.46 ± 1.02c | 9.80 ± 1.02a | 9.94 ± 1.02a |
| Oligomer formation[8] | $-DTT^5$ | No | Yes | No | No | No | No | No | No |
| | $+DTT^6$ | Yes | Yes | Yes | Yes | Yes | No | No | No |
| | Act.[7] | Yes | Yes | Yes | Yes | Yes | No | Yes | No |
| Toxicity (in nM) ($LC_{50}$ ± SE)[9] | $+DTT^6$ | 366.9 ± 1.9a | 293.4 ± 1.8a | 460.7 ± 1.6a | 301.9 ± 1.4a | 610.7 ± 1.6a | >30,000 | 3676.0 ± 1.5b | >30,000 |
| | Act.[7] | 84.7 ± 1.8a | 76.5 ± 2.0a | 94.2 ± 1.8a | 101.1 ± 1.9a | 105.9 ± 1.7a | >30,000 | 204.7 ± 1.3b | 1156.9 ± 1.1b |

**Fig. 3 Point mutations affect different steps along the Cyt1Aa bioactivation cascade.** For each row, different letters indicate significant differences between mutants. [1]Length and [2]width of crystals. [3]Productivity was visually estimated based on the quantity and aspect of the crystal-spore suspension collected. [4]$SP_{50}$ corresponds to the concentration at which 50% of crystals solubilize after 1 h incubation at RT. See Supplementary Table 2 for corresponding statistics. [5]Protoxin solubilized without DTT, releasing a disulfide-bridged dimer (except for C7S), and [6]with DTT, releasing a protoxin monomer. [7]Toxin monomer activated by proteinase K. [8]Capacity of each toxin species to generate a ladder pattern characteristic of membrane-bound oligomers (MBO) on 6% SDS-PAGE. See corresponding gels in Supplementary Fig. 9. [9]$LC_{50}$ corresponds to the concentration lethal for 50% of cell population. See Supplementary Table 2 for corresponding statistics. Scale bar = 0.2 μm. Source data are provided as a Source Data file.

mutation does not significantly affect pH sensitivity, eliminating the possibility of this residue partaking in the crystal dissolution mechanism (Fig. 3, Supplementary Table 2 and Supplementary Fig. 5). In contrast, the C7S crystals dissolve in the absence of DTT at the same pH as WT crystals soaked with DTT and remain unperturbed by addition of DTT (Figs. 3 and 4a). Hence, the disulfide bridge contributed by C7 increases the resilience of WT crystals by diminishing their pH sensitivity. Likewise, the Q168E mutation strongly increases pH sensitivity to lower pHs, with $SP_{50}$ of $10.0 \pm 1.0$ and $8.5 \pm 1.0$ in the absence and presence of DTT, respectively, confirming the suspected involvement of interface #3 (Fig. 2b, Supplementary Fig. 4 and Supplementary Table 1) in pH sensing and in the subsequent dissolution cascade (Fig. 3 and Supplementary Table 2). Irrespective of the presence of DTT, the D11N, E32Q and E45Q mutations do not significantly affect pH sensitivity (Fig. 3, Supplementary Table 2 and Supplementary Fig. 5), indicating that none of the pH-insensitive H-bonds introduced by mutation at the DS interface is on its own sufficient to significantly increase the $SP_{50}$.

We characterized the first active protoxin unit—i.e., that released upon crystal dissolution—by combined use of electrophoresis and mass spectrometry. By these two methods, we found that a dimer is predominantly released upon dissolution of crystals in the absence of DTT, accompanied by a monomer and by decreasing amounts of 3-mers, 4-mers and 5-mers (Fig. 4b and Supplementary Fig. 6). The dimer is disulfide-bridged at position C7, as demonstrated by the fact that it dissociates into monomers in presence of DTT and β-mercaptoethanol, but not upon heating to 95 °C, and by the observation of a single monomeric species upon dissolution of crystals of C7S, but not C190V (Fig. 4b and Supplementary Figs. 4–7). The observation of WT Cyt1Aa monomers upon dissolution of crystals in the absence of DTT nonetheless establishes that a fraction of the disulfide bridges break upon elevation of pH to 11 (Fig. 4b and Supplementary Fig. 6), consistent with the two alternate conformations observed in the "pH10" structure (Fig. 2b and Supplementary Fig. 4) and with the known pH sensitivity of disulfide bonds which can rupture upon pH elevation following a Cys-S-S-Cys + OH⁻ → Cys-S⁻ + Cys-S-OH oxidation reaction[16,17]. The pH-dependent

heat-stability profile of the Cyt1Aa protoxin dimer is also consistent with the presence of a disulfide bridge associating two protoxin monomers in the dimer (Supplementary Fig. 7)[18]. Residual dimers persist even at the highest concentrations of DTT and β-mercaptoethanol tested (200 and 715 mM, respectively), which likely are steady DS dimers (Supplementary Fig. 8). This hypothesis would explain the presence, upon dissolution of WT crystals in the absence of DTT (Fig. 4b and Supplementary Fig. 6), of higher order oligomers, corresponding to associations by disulfide bridges of a monomer and a DS dimer (3-mer), or of two DS dimers (4-mer), or of two DS dimers and a monomer (5-mer), etc. It would also rationalize the absence of these oligomers upon addition of reducing agents.

It remains uncertain whether or not the mosquito gut is a reducing environment[19] and thus under what form the protoxin is mainly released upon crystal dissolution in the natural context. We found that regardless of whether the protoxin is present in the form of a disulfide-bridged dimer or a protoxin monomer, addition of proteinase K, shown to faithfully mimic mosquito gut proteases[20], allows activation into a 23 kDa toxin monomer, consistent with the cleavage of its first 30 and last 5 amino acids[21] (Fig. 4b and Supplementary Fig. 6). Thus, irrespective of the redox state of the mosquito gut, a 23 kDa monomeric activated toxin is eventually made available upon activation by gut proteases. Nonetheless, two other Cyt1Aa species—a disulfide-bridged dimer (~54 kDa) and a protoxin monomer (~27 kDa)—could co-exist in the mosquito gut depending on its redox state (Fig. 4b and Supplementary Fig. 6).

We therefore investigated the propensity of these three Cyt1Aa species—the disulfide-bridged dimer, the protoxin monomer and the activated toxin monomer—and of their mutated variants to assemble into membrane-bound oligomers (MBO). When monomers of either the 27 kDa protoxin or the 23 kDa proteolytically activated toxin[7,21,22] are mixed with POPC liposomes (~100 nm radius) and electrophoresed on 6% SDS-PAGE gels, a ladder pattern characteristic of Cyt1Aa MBO is observed (Fig. 4c). The disulfide-bridged dimer is unable to form MBO upon contact with liposomes (~100 nm radius; LUVs) whereas the C7S mutant, able only to solubilize as protoxin monomers, forms MBO regardless of proteolytic activation (Fig. 3

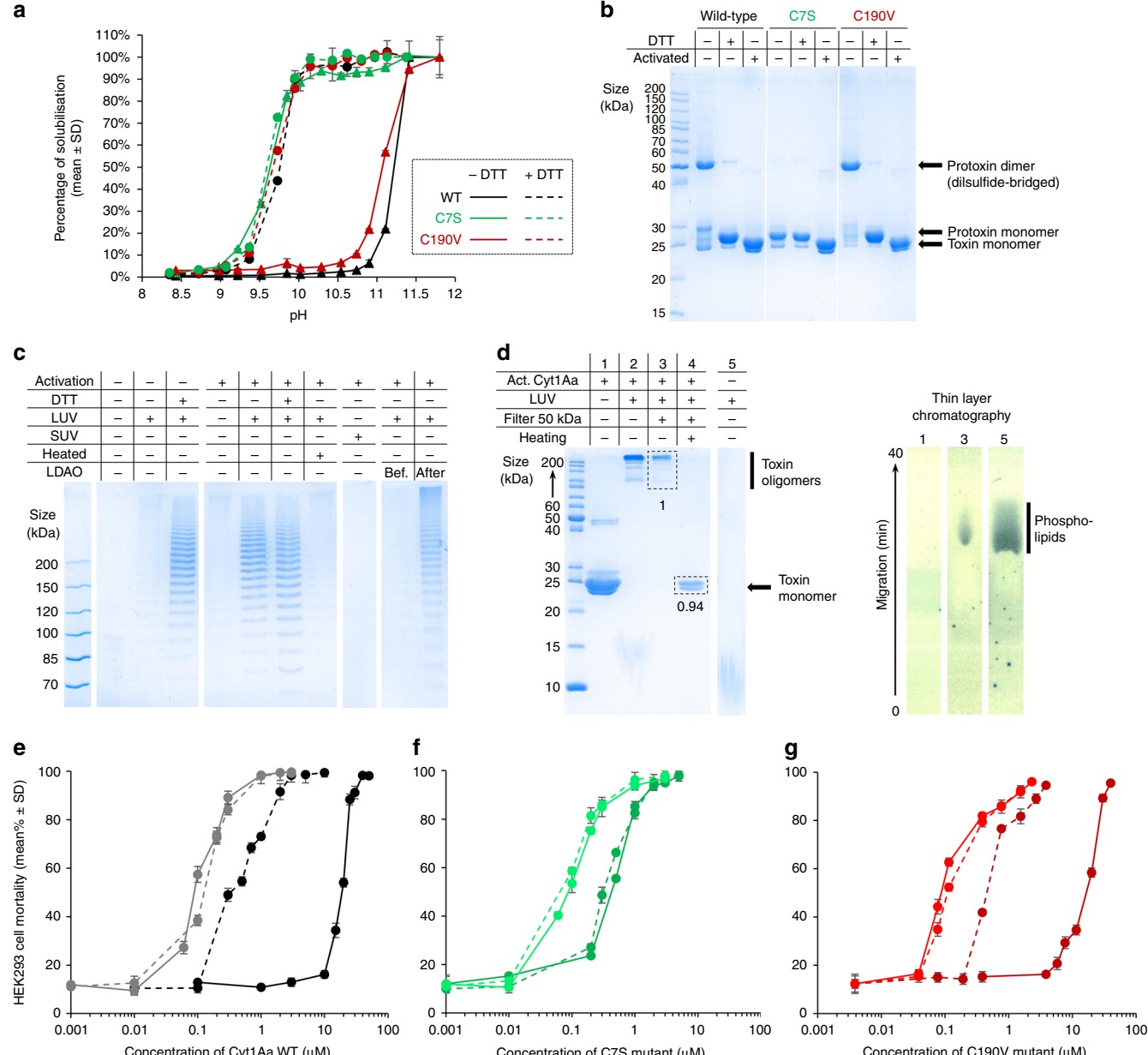

**Fig. 4 The disulfide bridge levels the pH sensitivity of crystals and abrogates toxicity. a** pH-dependent crystal dissolution. **b** Electrophoresis on a 12% SDS-PAGE gel supports that the wild-type (WT) dimer released upon dissolution of crystals in absence of DTT is disulfide-bridged at position C7. **c** Both the protoxin and toxin monomers can form membrane-bound oligomers (MBO) in presence of 100 nm radius liposomes, as revealed by a ladder-like profile on 6% SDS-PAGE gels. The disulfide-bridged dimer is unable to form MBO. **d** MBO are complexes of non-covalently bound Cyt1Aa monomers (as revealed by 15% SDS-PAGE of heated MBO) with phospholipids (as indicated by TLC). **e–g** Cytotoxicity of WT (**e** black), C7S (**f** green) and C190V (**g** red) Cyt1Aa was assayed against HEK293 cells in the three forms that may coexist upon dissolution of WT crystals, i.e. the disulfide-bridged protoxin dimer (solid dark lines; dissolution at pH 11.8 without DTT), the protoxin monomer (dashed dark lines; dissolution at pH 11.8 with DTT), and the activated toxin monomer in absence (solid light line) or presence (dashed light lines) of DTT. The WT activated toxin monomer is 4 and 200 times more active than the monomeric protoxin and the disulfide-bridged dimer, respectively. Error bars = SD. Source data are provided as a Source Data file.

and Supplementary Fig. 9). Thus the disulfide bridge, but not the N-terminal propeptide, can abrogate MBO formation. Thin layer chromatography (TLC) nonetheless indicates that the N-terminal propeptide masks half of the lipid binding interface in the oligomers, with protoxin and toxin MBO respectively featuring ~11.1 ± 2.6 ($N=4$) and ~22.1 ± 6.9 ($N=4$) phospholipids per Cyt1Aa monomer (Fig. 4d), consistent with previous estimations based on liposome disruption assays[23]. MBOs do not form upon contact of toxins with lipids solubilized in detergent micelles or bilayered in two-times smaller liposomes (~50 nm radius; SUVs)

(Fig. 4c), demonstrating a requirement for a fully formed membrane with a minimal radius of curvature. MBO persist after solubilization of their supporting liposomes and, despite an apparent spacing of bands by 11–17 kDa in SDS-PAGE gels (Supplementary Fig. 10), dissociate into full-size monomers upon heating to 95 °C (Fig. 4d and Supplementary Fig. 6). This observation shows not only that MBO formation entails drastic conformational changes after which Cyt1Aa may not solubilize as a monomer unless heated, but also that there is no further proteolytic cleavage occurring post-insertion into membrane nor

covalent link between monomers within MBO. MBO formation is unaffected by the C190V and Y171F mutations but is reduced in the E32Q protoxin and abrogated in the E45Q and Q168E protoxins, with full, partial and no restoration upon cleavage of the propeptides, respectively (Fig. 3 and Supplementary Fig. 9). This result indicates that stabilization of the N-terminal propeptide, and by extension of the DS dimer, inhibits MBO formation. That propeptide removal does not rescue MBO formation in E45Q and Q168E indicates that these mutations further impact processes downstream dissociation of dimers into monomers.

Treatment of Cyt1Aa MBO with detergents other than SDS (sodium dodecyl sulfate) prior to electrophoresis on native gels reveals the unique ability of SDS to produce a ladder pattern, effectively breaking down higher order MBO as they migrate through the gel (Supplementary Fig. 11). The smallest and largest MBO fragments migrate at the expected sizes for a Cyt1Aa trimer and 26-mer (~0.6 MDa; Fig. 4c), respectively, but cross-linking (using either glutaraldehyde or DTSSP (3,3′-dithiobis(sulfosuccinimidyl propionate)) thence affording 3 and 12 Å spacing between amines, respectively) prior to SDS-PAGE migration prevents MBO from entering gels, suggesting that they are natively larger (Supplementary Fig. 12). We note that similar ladder-like profiles, proposed to reflect formation of oligomers of non-fixed stoichiometry by stepwise addition of monomeric units[7] have been reported for the structurally homologous Cyt2Aa[24,25] and VVA2[26] toxins, as well as for pneumolysin[27], whose pore-forming domain resembles Cyt1Aa[28].

**Cytotoxicity originates from MBOs**. Cyt1Aa WT toxicity was assayed on insect Sf21 cells (0.04–0.4 μM) (Supplementary Fig. 13, Supplementary Movies 1 and 2) and on two mammalian cell lines, namely NIH fibroblast cells (0.04–4 μM) (Supplementary Fig. 13, Supplementary Movies 3 and 4) and HEK293 cells (0.001–30 μM) (Figs. 3 and 4e and Supplementary Table 2), supporting a generalist mode of action whose efficiency likely depends on the cell membrane phospholipid composition. HEK293 cells were chosen to pursue investigations, because their monodisperse nature enables use of a fluorescence-activated cell sorting (FACS) flow cytometer system, combined with propidium iodide (PI) staining as a reporter for cell death, to quantify the cytotoxicity of WT Cyt1Aa and mutants thereof. For each, $LC_{50}$ values were determined in the disulfide-bridged dimer (except C7S), protoxin monomer, and activated monomer forms (Fig. 3 and Supplementary Table 2).

All mutants and/or toxin forms capable of yielding MBO also display cytotoxicity (Figs. 3 and 4e–g and Supplementary Figs. 9 and 14), illuminating a direct link between MBO formation and cytotoxicity. Briefly, we found a four-fold increased toxicity for the activated toxin ($LC_{50} = 0.085$ μM) when compared to the monomeric protoxin ($LC_{50} = 0.36$ μM), which could stem from the reduced interaction surface for phospholipids on the protoxin (Figs. 3 and 4e). Contrastingly, the disulfide-bridged dimer exhibits a ~200 fold lower toxicity ($LC_{50} = 16.6$ μM) (Figs. 3 and 4e) which, given the inability of this dimer to form visible MBO on SDS-PAGE gels (Fig. 4c), could be due to residual monomers present in the sample (Fig. 4b and Supplementary Fig. 6). The C7S, C190V and Y171F mutants, which form MBO, display the same toxicity as the WT (Fig. 3, Supplementary Fig. 14 and Supplementary Table 2), excluding a possible role for these residues in cytolytic oligomer formation. Addition of DTT to proteolytically activated WT, C7S and C190V Cyt1Aa does not significantly change cytotoxicity ($LC_{50} = 0.112$ μM), further demonstrating that neither cysteine is involved in the formation of Cyt1Aa cytolytic oligomers and eliminating a DTT effect on toxicity (Figs. 3 and 4e–g and Supplementary Table 2). In

contrast, cytotoxicity is affected in all but one of the DS and interface #3 mutants, mirroring their inability or reduced ability to form MBO. The E32Q mutant shows a 10 and 2.4 fold reduced toxicity in the protoxin and toxin forms compared to WT, respectively, suggesting that the mutation inhibits the transition into an MBO conformer (Fig. 3 and Supplementary Fig. 14). The E45Q mutant shows no toxicity in the protoxin form (up to 30 μM) and a 14-fold reduced toxicity upon proteolytic activation, indicating that in addition to inhibition of the transition into an MBO conformer due to stabilization of the DS dimer by H-bonding to E32, the pH-desensitization of the β2 (E45Q) tether to the αC/αD hairpin (Q138) blocks another important step in the formation of cytolytic oligomers (Fig. 3 and Supplementary Fig. 14). Most strikingly, the Q168E mutation eliminates cytotoxicity of both the protoxin and the activated toxin (up to 30 μM), again paralleling the inability of these to form MBO and suggestive of Q168 being involved in the interaction with—and possibly the perforation of—cell membranes (Fig. 3 and Supplementary Fig. 14). Interestingly, the reverse mutation D11N does not significantly affect toxicity, clearly splitting the different roles played by Q168 in crystal formation and dissolution, and in cytotoxicity (Fig. 3 and Supplementary Fig. 14).

**Porous oligomers fully perforate cell membranes**. How Cyt1Aa exerts its cytolytic activity remains unclear. It has been proposed, based on electrophysiology data, that Cyt1Aa oligomerizes into cation-selective channels of 6–20 Å diameter[5], but we failed to observe the formation of such pores despite conducting similar black lipid membrane (BLM) experiments at Cyt1Aa concentrations ranging from 1.7 to 60 μg mL$^{-1}$ (Fig. 5a). Rather, our observations are suggestive of a cooperative membrane-binding process, whereby multiple toxins successively insert and co-aggregate in the bilayer (inducing flickering in the measured electrical current) before the latter is ripped apart[3]. We therefore further challenged the porous nature of MBO by exposing Sf21 (insect) (Fig. 5b) and NIH 3T3 (mammal) cells (Fig. 5c) simultaneously to Cyt1Aa toxin at sub-lethal concentration and to fluorescent dextran beads of 1.4–8.5 nm. We found that beads up to 8.5 nm can penetrate cells through Cyt1Aa-induced lesions, excluding the possibility that Cyt1Aa forms a selective pore.

We used AFM on supported lipid bilayers (SLB) to obtain further insights into the large-scale structural dynamics of Cyt1Aa MBO (Fig. 5d and Supplementary Fig. 15). Minutes after the addition of Cyt1Aa WT activated toxin, a first type of MBO—hereafter referred to as membrane-bound aggregates (MBA)—forms from the continuous encounter of monomers freely diffusing at the membrane surface (Fig. 5d–f), with a strong significant positive correlation between the time elapsed since toxin addition and the surface area occupied by the MBA (Fig. 5g). Toxin aggregation eventually leads to the formation of holes at the periphery of MBA (Fig. 5h, i), consistent with cell microscopy assays (Fig. 5b, c).

We monitored the area occupied by MBA and holes as a function of hole depth. The significant negative correlation observed between the hole depth and MBA surface (Fig. 5j), but not between the hole depth and the total "MBA+hole" surface (Fig. 5k), suggests that two types of MBO coexist: porous oligomers and MBA. Full membrane perforation is visible only for holes of ~54 nm diameter, with a ~2300 nm² surface and ~169 nm circumference, corresponding to a porous oligomer formed by the assembly of ~56 monomers (Fig. 5j). That only a fraction of the porous oligomers displays sufficient depth to fully perforate the membrane (~4.5 nm) suggests that the structural transition between MBA and porous oligomers is independent of

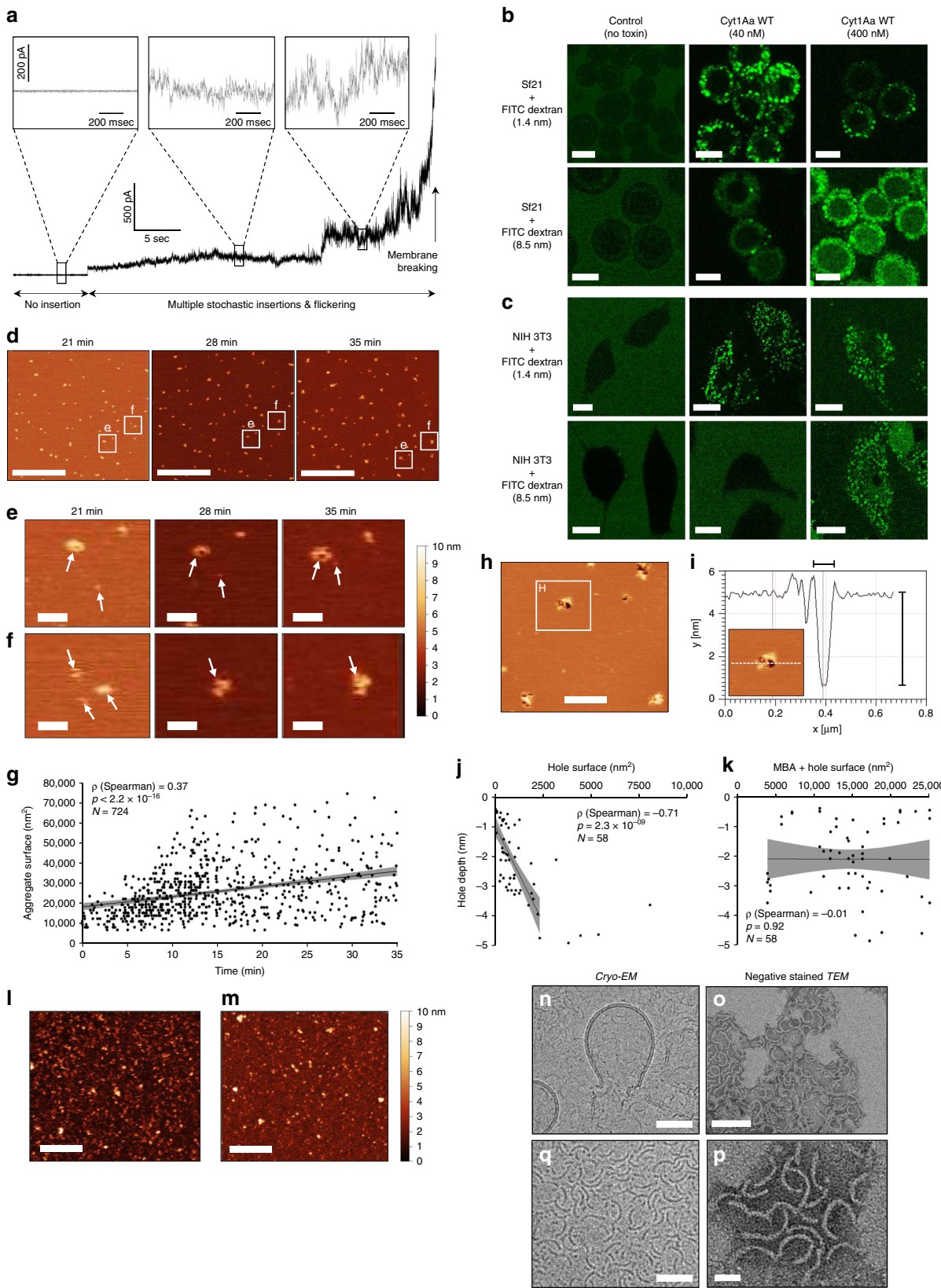

MBA size and could involve formation of a pre-pore (Fig. 5j). Experiments on the Q168E mutant reveal that this mutation affects the first step of toxin insertion into the membrane. Indeed, the Q168E mutant is unable to penetrate the membrane and aggregate to form pores (Fig. 5l), behaving like the BSA control (Fig. 5m).

We attempted to characterize the porous oligomers by TEM, either after negative staining or under cryogenic conditions (cryo-EM). Addition of Cyt1Aa to 100 nm liposomes led to their almost total disruption, with the few remaining LUVs exhibiting membrane leakage (Fig. 5n), and to the release of a homogenous population of 3.50 ± 0.42 nm thick and 30.3 ± 2.3 nm long

**Fig. 5 Cyt1Aa forms oligomers that fully perforate and eventually disrupt lipid bilayers. a** Single cation-channel formation was not observed in black lipid membrane (BLM) experiments. **b**, **c** Cyt1Aa allows entry in both Sf21 (**b**) and NIH 3T3 cells (**c**) of co-incubated FITC-labelled (fluorescein isothiocyanate) dextran beads up to 8.5 nm in size. Scale bars = 10 μm. **d**–**f** Membrane-bound Cyt1Aa monomers exhibit mobility (**d**, **e**) and display the capacity to merge into larger membrane-bound aggregates (MBA). Scale bars = 3 μm (**d**) and 300 nm (**e**, **f**). **g** A significant positive correlation was observed between the surface of MBA and the time elapsed since toxin addition. **h** 35 min after toxin addition, membrane perforation is observed at the periphery of MBA. Scale bar = 500 nm. **i** The depth of holes can reach 4.5 nm, consistent with a full spanning of the lipid bilayer. **j**, **k** A significant negative correlation is observed between the surface of holes and their depth (**j**), but not between the latter and the combined area of the hole and the parent MBA (**k**). **l**, **m** Formation of MBA and holes was neither observed for the activated form of the non-toxic Q168E mutant (**l**) nor for the BSA control (**m**). Scale bars = 500 nm. **n**–**q** Transmission electron microscopy captures liposome lysis by the toxin (**n**), and the resulting release of arciform oligomers (**o**–**q**). Scale bars = 50 nm (**n**, **q**), 20 nm (**p**) and 100 nm (**o**). Source data are provided as a Source Data file.

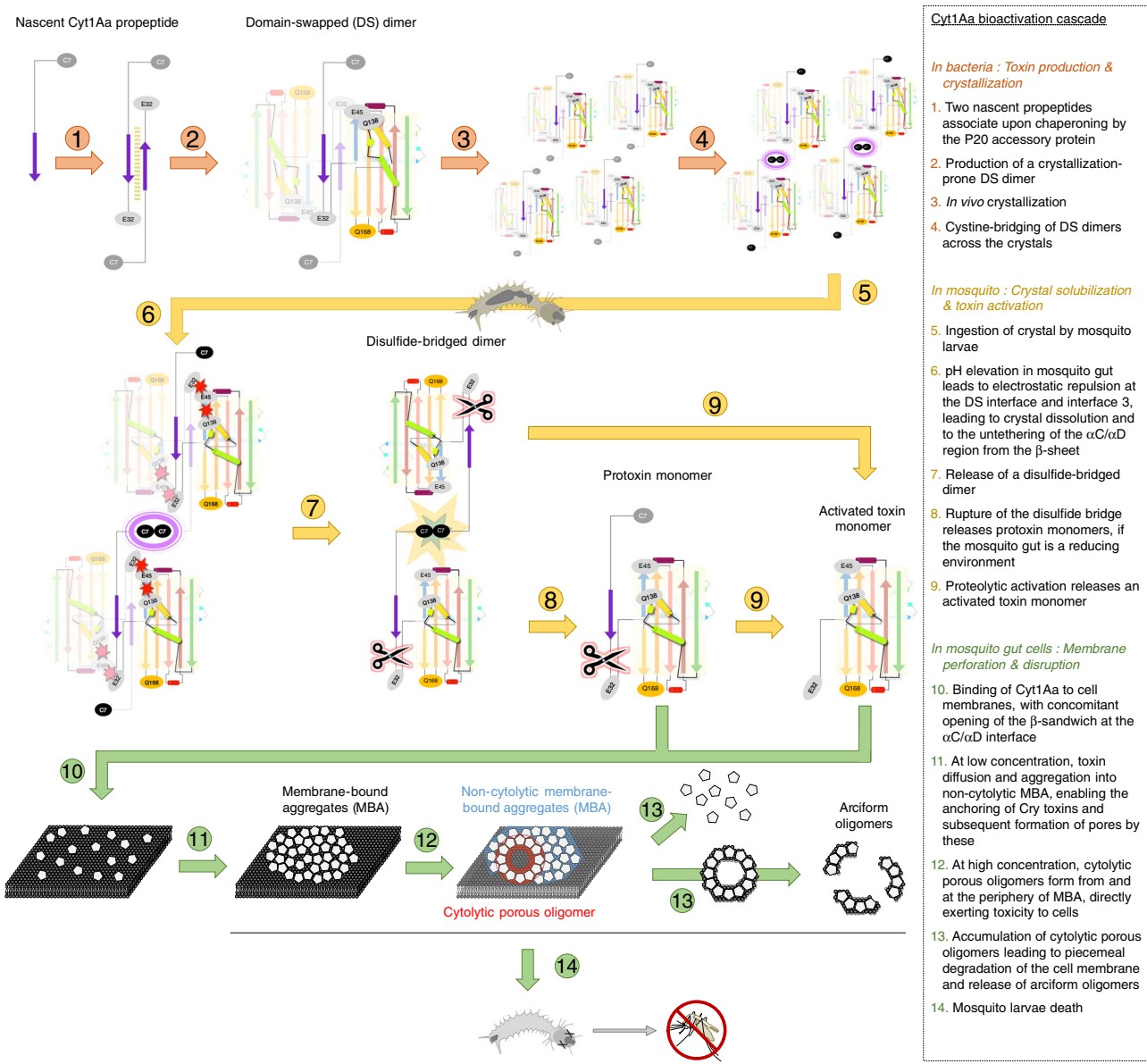

**Fig. 6 Proposed model for Cyt1Aa bioactivation cascade.** The bioactivation cascade of Cyt1Aa starts with dimerization through a domain-swapped interface, which allows both self-inhibition and in vivo crystallization, and ends with oligomer formation in the membrane of mosquito gut cells. The Cyt1Aa structure is abstracted and colored sequence-wise, from cold (N-terminus) to hot (C-terminus) colors. The magenta square highlights the disulfide bridge between domain-swapped (DS) dimers. Red starbursts indicate electrostatic repulsion, whereas the yellow-blue starburst indicates disulfide bridge disruption. Conformational changes occur in the toxin upon pH elevation, resulting in an untethering of the αC/αD hairpin from the β-sheet. We propose that upon contact with a cell membrane, the protein fully opens at the αC/αD hydrophobic interface and that the two thereafter exposed hydrophobic surfaces appose onto the membrane bilayer, yielding the membrane-bound aggregate (MBA) conformer. Aggregation of MBA conformers eventually results in the formation of holes, at the periphery of MBA, resulting in the death of midgut cells.

arciform oligomers, characterized by a mean curvature of 129.3 ± 7.4° (mean ± SD; N = 27) (Fig. 5n–q). These could represent either the open form of a porous oligomer or pieces thereof. In the first case scenario, i.e. assuming conservation of the length of arciform oligomers, porous oligomers would be ~12 mers of ~276 kDa characterized by a 2–5 nm pore, which is inconsistent with our native and SDS-PAGE, AFM and BLM conclusions. Thus, we favor the hypothesis that arciform oligomers represent pieces from the breakdown of larger cytolytic oligomers, such as those observed by AFM.

## Discussion

Structural investigations of in vivo-grown sub-micron-sized crystals using SFX, AFM, and complementary methods have shed light on key steps in the Cyt1Aa bioactivation cascade, from in vivo crystallization in *Bti* cells, to crystal dissolution, proteolytic activation, and membrane insertion and perforation through oligomerization (Fig. 6). Our SFX structures point to the N-terminal propeptide of Cyt1Aa being a key structural element which, by dimerization through a DS interface[13,14], intervenes in Cyt1Aa folding, self-inhibition and in vivo crystallization—but not in crystal dissolution. Rather, we unveil the respective roles played by the disulfide-bridge and interface #3, showing how these act together to ensure crystal dissolution occurs only under highly alkaline (and possibly reducing) conditions, such as those found in the larval mosquito midgut. This mechanism differs slightly from that highlighted in naturally occurring nanocrystals of BinAB, wherein no cysteine interface is in play and it is rather pH-induced electrostatic repulsion at crystal interfaces between tyrosine residues and their obligate H-bond acceptors that complements the analogous mechanism between acidic side chains[29]. Further structural characterization of naturally crystalline protoxin structures may allow the discovery—and eventually the utilization—of additional mechanisms of in vivo crystallization and controlled pH-dependent solubilization.

It remains unclear whether the mosquito gut is a reducing environment[19]. If so, crystals could dissolve and begin to initiate cytopathology at a pH as low as 9.5, in line with the observation of Cyt1Aa lesions to anterior midgut cells[30]; if not, a disulfide-bridged dimer would be released upon crystal dissolution, which would need to be proteolytically activated. We showed that both this dimer and the protoxin monomer can be processed by proteases, yielding the 23 kDa activated toxin. We also show that upon contact with a lipid membrane, drastic conformational changes take place, most likely due to an opening of the structure at the αC/αD hydrophobic interface with the β-sheet. This interface indeed demonstrates the greatest structural differences observed between the protoxin at pH 7, the protoxin at pH 10 and the activated toxin at pH 8 (Figs. 1 and 2, and Supplementary Figs. 2–4). The hypothesis that an opening of the protein at the αC/αD hydrophobic interface is required for MBO formation was examined by the introduction of a E45Q mutation, intended to render pH-insensitive the tether between the β-sheet and αD (Q138(OE1)), besides stabilizing the DS dimer. The mutation favoured crystal growth but resulted in a 14-fold reduced toxicity for the activated toxin, while eliminating toxicity of the protoxin (Fig. 3). It thus seems plausible that opening of the WT structure at the αC/αD hydrophobic interface with the β-sheet is an important step for membrane insertion and subsequent MBO formation. That the αC/αD hairpin would be involved in MBA formation is in line with its burial at the DS interface in the protoxin structure.

Evidence was also provided that Q168, buried at crystal packing interface #3, is a key player in crystal formation (markedly different crystal dimensions and production yield upon

Q168E mutation), pH-induced dissolution (~1 pH unit difference upon Q168E mutation) and toxic activity (fully abrogated in the Q168E mutant) (Fig. 3). AFM imaging and SDS-PAGE analyses indicate that the mutation produces effects at the membrane insertion level, which we interpret as an indication that this residue, at the tip of β5, is part of the β-sheet structure that plunges into the membrane to form the porous oligomer. This proposition is in agreement with earlier mass spectrometric determination of the extent of membrane-inserted domains in the Cyt1Aa porous oligomers[31]. These studies indeed pointed to residues 42 to 132 forming the hydrophilic part of the pore (β2-αA-αB-β3), and to residues 154 to 234 as being membrane-associated and thus forming the hydrophobic part of the pore. These are all residues located between the C-terminus of αD and the N-terminus of αF (Fig. 1), namely the αD-β4 loop and the β4-β5-β6-β7 β-sheet featuring the short αE helix between β6 and β7. It was also shown that the N- and C-termini of the protein are both exposed on the extracellular side of the pore[31,32]. The simplest way to reconcile these two results is to envision a pore wherein β4 (and possibly the αD-β4 loop) contributes one of the strands of the oligomerizing β-sheet structure, with the β4-β5 loop pointing on the intracellular side—in line with our proposal that Q168 is part of the β-sheet structure that plunges into the membrane. Regardless, the absence of β2 and β3 from the membrane-inserted segment of the pore[31] indicates that these must respectively dissociate from β4 and β7 on each edge of the β-sheet, to enable transition to the porous oligomer. It remains to be determined if this step is completed at the monomer to MBA-conformer (i.e. upon interaction with the membrane) or at the MBA-conformer to porous oligomer transition (i.e. upon side-by-side contact between MBA conformers).

The exact structures of the Cyt1Aa MBA conformers and porous oligomers remain to be determined. Our CryoEM results revealed arciform oligomers from which a high resolution structure could not be derived. CryoEM investigations on pneumolysin[33], whose pore-forming domain resembles Cyt1Aa[28] and which alike Cyt1Aa yields ladder-like profile on SDS-PAGE gels[27] and arciform oligomers upon insertion into liposomes[34,35], suggest that a prerequisite to obtaining a high resolution cryoEM structure of Cyt1Aa porous oligomers will be to carefully devise a lipid/detergent/additive formulation capable of stabilizing their annular architecture, i.e. preventing their fall-off into arciform oligomers. At present, we can nonetheless propose a model that fits all the information developed by us and others on Cyt1Aa pore formation[2–6,21,27,31,36] (Supplementary Fig. 16). Our AFM, BLM, TEM and in vivo cell assay data exclude the possibility that Cyt1Aa exerts toxicity by forming a cation-channel of 6–20 Å diameter (the "pore-forming" model)[4–6], or by acting as a detergent with molecules bound to—but not inserted into—the membrane, (the "detergent-like" model)[2,3]. Rather, the data taken together suggest that following the untethering of the αC/αD hairpin from the β-sheet due to pH elevation and proteolytic activation, the Cyt1Aa structure opens at this locus upon membrane contact, yielding a new membrane-bound Cyt1Aa conformer—the MBA conformer. Side-by-side contact between the β-sheets of adjacent MBA protomers would enable the observed transition to the porous conformation. Our data show that this cooperative MBA-to-porous conformer transition occurs only after a critical number of molecules is recruited, yielding the observed >54 nm diameter membrane perforation upon concomitant plunging across the bilayer of the β-sheets of ~56 or more associated porous conformers (Figs. 5 and 6)—possibly with their β4-β5 loops and αE helices pointing towards the intracellular side[31,36]. Importantly, this proposed model reconciles the data hitherto presented to oppose the "pore-forming" and "detergent-like" models[2,3], e.g. explaining how the N-terminal

and C-terminal parts of the toxin can both be exposed on the outer face of the membrane[31,36]; how can the membrane-inserted segment of porous oligomers feature Cyt1Aa residues spanning αD to αF[31,36]; how the large pores formed by Cyt1Aa can enable transit into midgut cells of large molecules or proteins, such as the 42 kDa BinA toxin[30]; how the non-porous Cyt1Aa oligomers, here identified as MBA, may serve as substitutes for mosquito receptors of Cry toxins, enabling them to dock on mosquito midgut cell membranes and subsequently assemble into toxic pores even in absence of their specific membrane-bound receptors. Lack of expression of toxin receptors, or expression of defective or soluble variants thereof, is indeed one of the most efficient mechanisms developed by insects to resist *Bt* toxins[37,38]. We admittedly leave open the question of the exact structure of Cyt1Aa porous oligomers, with hope that high resolution cryoEM structures of the Cyt1Aa porous oligomer and/or MBA conformers may soon shed further light on the issue.

Besides clarifying our understanding of the Cyt1Aa bioactivation cascade, our study provides insights into how the yield (D11N, E32Q), toxic function (E45Q, Q168E) and pH sensitivity of Cyt1Aa crystals (C7S, Q168E) can be curbed by single atom substitutions, opening avenues to exploit recombinant Cyt1Aa variants as improved mosquitocides with increased production yield, extended target spectrum and improved toxicity. Results furthermore push SFX and our difference-density based mutation strategy forward as means to obtain structural and functional insights into naturally crystalline insecticidal proteins.

## Methods

**Plasmid construction, crystal production, and purification.** The shuttle vector pWF45 was used to produce in vivo crystals of wild-type Cyt1Aa toxin[8,39]. It was also used as a backbone to construct plasmids containing single-point mutated *cyt1aa* genes. A total of 7 point-mutants of Cyt1Aa were constructed based on the difference-density maps generated from crystallographic data (Fig. 2 and Supplementary Fig. 4). A detailed description of the mutation strategy is available in Supplementary Note 1. Our extensive analysis of the bibliography indicates that none of these mutants has been constructed in previous studies (Supplementary Table 3 and Supplementary Fig. 16).

Point mutations were inserted into *cyt1aa* gene sequence by Gibson assembly (the list of primers used for plasmid construction is available in Supplementary Table 4). For each mutant, two fragments were amplified from pWF45 using two different primer couples. The fragments were complementary by their 15–18 bp overlapping 3′ and 5′ overhangs with a target Tm of 50 °C. The point mutation was inserted into the overhangs of the two fragments spanning the *cyt1aa* gene. For each mutant, the two fragments were assembled using the NEBuilder HiFi DNA Assembly (New England BioLabs) by following manufacturer's instructions. After 90 min of incubation at 50 °C, the constructed plasmids were transformed by the heat shock procedure into chimiocompetent Top10 *Escherichia coli* strain (New England BioLabs). Colonies were selected on LB agar medium supplemented with ampicillin (100 µg mL⁻¹) and plasmids were extracted by using the NucleoSpin Plasmid extraction kit (Macherey-Nagel). Successful plasmid constructions were validated by double digestion (EcoRI and BamHI) followed by migration on 1% agarose gel stained with SYBR Safe (Invitrogen) and presence of mutations was assessed by Sanger sequencing at the Eurofins Genomics sequencing platform.

Validated plasmids were transformed into the acrystalliferous strain 4Q7 of *Bacillus thuringiensis* subsp. *israelensis* (*Bti*; The Bacillus Genetic Stock Center (BGSC), Columbus OH, USA) with an improved electrotransformation procedure[40] using a MicroPulser Electroporator (BioRad). Colonies were selected on LB agar medium supplemented with erythromycin (25 µg mL⁻¹) and used to inoculate an overnight 5 mL LB liquid preculture. Precultures were spread on T3 sporulation medium and incubated at 30 °C for 4 days to promote *Bti* sporulation and toxin crystal production. Spores and crystals were collected using a cell scraper, resuspended in water and centrifuged once at 10,000 *g* for 45 min. The pellet was resuspended in water and purified using a discontinuous sucrose gradient (67-72-79%). After 16h of ultracentrifugation at 23,000 *g* and 4 °C, the crystals were recovered from the 67–72% interface. Several rounds of centrifugation and resuspension in water were performed to discard as much sucrose as possible. Crystal purity was verified by SDS-PAGE on 12% gels. Purified crystals were conserved in ultrapure water at 4 °C until use.

**Crystal visualization by SEM.** For crystal visualization by SEM, crystals were resuspended in a 25 mM ammonium acetate solution, deposited on microscope circular glass slides and left for drying under a Sorbonne hood for 1 h or more. Samples were coated with a 2 nm thick gold layer with the Leica EM

ACE600 sputter coater and imaged using the Zeiss LEO 1530 scanning electron microscope (SEM). From exported TIFF images, the length and width of crystals of Cyt1Aa WT and all mutants were measured using the software ImageJ v1.51k (*N* = 40 crystals)[41]. Normality of data was confirmed by a Shapiro–Wilk test. Differences in length, width and length/width between Cyt1Aa WT and the different mutants were tested with an ANOVA test followed by a post-hoc Tukey HSD test performed using software R v3.5.2[42].

**Crystal visualization by AFM.** For crystal visualization by AFM, 5 µL of purified Cyt1Aa crystals conserved in ultrapure water were deposited on freshly cleaved mica without any further treatments and deposited in a desiccation cabinet (Superdry cabinet, 4% relative humidity) for 30 min. Imaging was performed on a Multimode 8, Nanoscope V (Bruker) controlled by the NanoScope software (Bruker, Santa Barbara, CA). Imaging was done in the tapping mode (TAP) with a target amplitude of 500 mV (about 12 nm oscillation) and a variable setpoint usually around 70% amplitude attenuation. TESPA-V2 cantilevers (k = 42 N m⁻¹, Fq = 320 kHz, nominal tip radius = 7 nm, Bruker probes, Camarillo, CA, USA) were used and images were collected at ~1 Hz rate, with 512 or 1024 pixel sampling. Images were processed with Gwyddion[43], and if needed stripe noise was removed using DeStripe[44].

**Crystal preparation and injection via MESH-on-a-stick.** SFX experiments were performed at the Coherent X-ray Imaging (CXI) instrument of the Linac Coherent Light Source (LCLS) at the SLAC National Accelerator Laboratory of Stanford University (California, USA). Crystals of Wild-Type (WT) Cyt1Aa ("pH7" dataset) and of the C7S mutant ("C7S" dataset) were suspended into spin buffer (MES 0.1M, NaCl 0.1M, Glycerol 50%, pH 6.5) and delivered across the X-ray beam at a concentration of 1.6% (grams of crystals in 100 mL of buffer solution) in the SCC sample-chamber at room temperature and under vacuum, using the microfluidic electrokinetic sample holder (MESH) method, described more fully in Sierra et al.[11]. Cyt1Aa WT crystals were also injected in (i) spin buffer supplemented with 1 mM of freshly prepared DTT, to investigate the structural effects of disruption of disulfide bonds between N-terminal propeptides—"DTT" dataset; and in (ii) an alkaline spin buffer (CAPS 0.1M, NaCl 0.1M, Glycerol 50%, pH 10.3) for 6 h 30 min prior to injection to characterize the structural changes that drive crystal solubilization at high pH—"pH10" dataset. These redox and pH conditions were chosen after verification that crystals do not dissolve on the timescale of hours.

Specific to MESH injection, a continuous 1.5 m long polyamide-coated fused-silica capillary of 100 µm inner diameter and 360 µm outer diameter was used to deliver the sample into the SCC vacuum chamber. Approximately 800 µL of sample slurry with glycerol additive was pipetted in a microcentrifuge tube, which was then placed in a small pressurized sample holder. The capillary and the platinum wire used as an electrode were fed through the pressure cell and immersed in the slurry. A low backing pressure of 5 psi nitrogen gas was applied in the sample holder to aid the injection. The voltage was applied by a Stanford Research Systems PS350 (Sunnyvale, CA) high voltage source and was held between 4300 and 4500 V (currents < 1 µA) while the counter electrode was grounded. The flow rate was not directly measured, but we estimate that the sample consumption was approximately 2 µL min⁻¹, as judged from crude measurements of leftover sample volume. The four structures presented herein were collected in less than 12 h of continuous beamtime and consumed less than a milliliter of sedimented crystals.

**Data collection and processing, and structure refinement.** Datasets were collected with a XFEL beam focused to a 1.3 µm FWHM spot and characterized by a wavelength of 1.28 Å (proposals P125 and P141). The sample chamber was at room temperature, under vacuum. We attempted to index all collected images with DIALS[45], using the *cctbx.xfel* graphical user interface[46,47]. The final "pH7", "DTT", "pH10" and "C7S" datasets consisted of 8462, 18766, 19924, and 7683 indexed patterns. Data were merged using *cxi.merge*[48], with negative intensities included, and resolution cut-offs were determined based on completeness (>99%), redundancy (>60) and CC₁/₂ (>0.05); in all datasets, <I/sigI> in the highest resolution shells are greater than 2.8. We phased the pH7 dataset by molecular replacement with Phaser, using as a starting model the activated structure of Cyt1Aa ("3ron [10.2210/pdb3RON/pdb]"[12]). Missing propeptide residues were manually rebuilt based on the electron density. The three other structures were phased by rigid-body refinement, using as a starting model the pH7 structure. Further refinement consisted in iterative cycles of manual rebuilding in real space using Coot[49], and refinement in the reciprocal space using Refmac[50]. In the final structures, the percentage of residues in the favoured, allowed and outlier regions of the Ramachandran plot and the clash-score and Molprobity scores are 98.35, 1.65, 0.0, 3.89, and 1.37 for the "pH7" structure (6T14; [10.2210/pdb6T14/pdb]); 98.75, 1.24, 0.0, 5.38, and 1.67 for the "DTT" structure (6T19; [10.2210/pdb6T19/pdb]); 98.35, 1.65, 0.0, 4.51, and 1.75 for the "pH10" structure (6T1A; [10.2210/pdb6T1A/pdb]); and 99.17, 0.83, 0.0, 2.05, and 1.36 for the "C7S" structure (6T1A; [10.2210/pdb6T1C/pdb]). Data collection, processing, and refinement statistics are shown in Table 1. We obtained experimental insights into pH and redox induced conformational changes by calculating structure factor amplitude Fourier difference maps (Fo–Fo) between the "pH7", "DTT", "pH10" and "C7S" datasets. To improve the estimate of structure factor amplitude differences, Fo–Fo maps were q-weighted

as described[51] and produced using a CNS[52] custom-written script[53]. Application of the q-weighting scheme to the diffraction datasets was essential to eliminate noise and amplify the difference signal. Fourier difference maps and difference distance matrix (DDM) calculations were performed using custom-written scripts.

**Crystal solubilization assays**. To assess the stability of the crystals formed by Cyt1Aa WT and its different mutants, we determined at which pH the crystals solubilize (Figs. 3 and 4a and Supplementary Fig. 5). Crystal suspensions were centrifuged at 11,000 $g$ for 2 min and the pellets were resuspended in 50 µL of 0.1 M of $Na_2CO_3$–$NaHCO_3$ solutions buffering at pH ranging from 9.0 to 11.8, or of $Na_2HPO_4$-$NaH_2PO_4$ solutions buffering at pH ranging from 8.5 to 9.0, following guidelines from the Sigma Aldrich Buffer Reference Center. Crystals were exposed for 1 h at RT in duplicates at each pH and then centrifuged at 11,000 $g$ for 2 min. The supernatant was promptly and carefully collected. The concentration of solubilized toxin was quantified using a Nanodrop 2000 (Thermo Fisher Scientist) by measuring the OD at 280 nm and by using the molar extinction coefficient and toxin size (27055 $M^{-1}$ $cm^{-1}$ and 27.3 kDa, respectively, as calculated with the ProtParam tool of ExPASy (https://www.expasy.org) using the Cyt1Aa protein sequence available under accession number "Q7AL78 [https://www.uniprot.org/uniprot/Q7AL78]"). Data are presented as the percentage of solubilization, calculated by dividing the protein concentration measured at a given pH by that measured at pH 11.8. For statistics, the best fitted model for the data was selected among four logistic regression models for binomial distribution (logit, probit, complementary log–log transformation (cloglog) and Cauchy distribution) by comparing their deviance using a script modified from[54] to calculate for each toxin the $SP_{50}$ (pH leading to solubilization of 50% of crystals). 95% confidence intervals (CI) were calculated with a Pearson's chi square goodness-of-fit test implemented in the software R 3.5.2[42,54]. Differences in $SP_{50}$ between mutants were considered significant when 95% CI did not overlap[55].

**Determination of the different forms of Cyt1Aa**. We investigated the three Cyt1Aa species possibly formed, in the mosquito larvae gut, upon dissolution of crystals and dissociation at the DS interface (namely, the disulfide-bridged dimer), possible reduction of disulfide bridges (namely the protoxin monomer) and activation by proteolysis (namely the activated monomer) (Fig. 4b, Supplementary Fig. 6). Crystals of WT Cyt1Aa and of all mutants were solubilized for 1h at RT in 0.1 M $Na_2CO_3$ buffer pH 11.8 and then were centrifuged at 11,000 $g$ for 1 min. The supernatant contained the solubilized protoxins (condition 1). Addition of DTT allowed the breakage of disulfide bonds between protoxin units, if any (condition 2). Soluble protoxins were activated into toxins by the addition of proteinase K or trypsin (condition 3). For further experiments, toxin was activated by proteinase K rather than other enzymes, such as trypsin, as it was shown to better mimics the activation performed by gut enzymes in insects[20,21]. The different forms of Cyt1Aa were analyzed by two methods: SDS-PAGE and MALDI-ToF.

For SDS-PAGE experiments, unheated samples were electrophoresed on 12% SDS-PAGE gels after addition of Laemmli buffer devoid of DTT. After staining by overnight incubation in InstantBlue (Sigma Aldrich, France), gels were washed twice in ultrapure water and migration results were digitalized using a ChemiDoc XRS+ imaging system controlled by Image Lab software version 6.0.0 (BioRad, France).

MALDI-ToF mass spectra were acquired on an Autoflex mass spectrometer (Bruker Daltonics, Bremen, Germany) operated in linear positive ion mode. External mass calibration of the instrument, for the 10–70 kDa m/z (mass/charge) range, was carried out using a protein calibration standard II from Bruker Daltonics. All samples were prepared as described above, except for experiments carried out directly on Cyt1Aa crystals, which involved dissolution of crystals in a 70:30 acetonitrile/water mixture. All Cyt1Aa samples were mixed in variable ratios (1:5, 1:10 or 1:20, v:v) with sinapinic acid matrix (Sigma; 20 mg mL$^{-1}$ in water/acetonitrile/trifluoroacetic acid, 70/30/0.1, v/v/v) to obtain the best signal-to-noise ratio spectra. 1–2 µL of the mixture was deposited on the target and allowed to air dry. Mass spectra were acquired in the 10–160 kDa m/z range and data processed with the Flexanalysis software (v.3.0, Bruker Daltonics).

**Heat stability of protoxin**. The stability of WT Cyt1Aa protoxin dimers was assayed at different pH and increasing temperatures (Supplementary Fig. 7). Crystals of Cyt1Aa WT were solubilized in 0.1 M $Na_2CO_3$ buffer pH 11.8, centrifuged and the supernatant containing soluble protoxin dimers was collected. Suspensions were diluted 50 times in buffers at different pHs. $Na_2CO_3$–$NaHCO_3$ buffer solutions (0.1 M) were used for pH 9, 10, and 11, and $Na_2HPO_4$–$NaH_2PO_4$ buffer solutions, for pH 7, 8, and 9, following guidelines from the Sigma Aldrich Buffer Reference Center. Complete buffer exchange was afforded by two steps of concentration using a AMICON ultra-filtration unit. Briefly, the protoxin suspension at pH 11 was concentrated 50 times using a 10 kDa cutoff (Sigma Aldrich, France), and then diluted 50 times in the buffer of interest (pH 7, 8, 9, or 10); this two-step procedure was repeated twice to achieve a complete buffer exchange. Suspensions were then added to Laemmli buffer devoid of DTT. To test the stability of the disulfide-linked dimer, each sample at each pH was heated for 5 min at 95, 100, 105, 110, 115, 120, 125, or 130 °C prior to loading on a SDS-PAGE 12% gel. Each temperature was tested in triplicate. Gels were stained by overnight

incubation in InstantBlue (Sigma Aldrich, France), washed twice in ultrapure water and digitalized using a ChemiDoc XRS+ imaging system controlled by the Image Lab software version 6.0.0 (BioRad, France). The relative intensity of the bands corresponding to Cyt1Aa dimers was estimated using the software ImageJ v1.51k[41].

**Insect and mammalian cells**. The effect of the Cyt1Aa toxin was tested on three different cell lines, namely an insect cell line (Sf21 cell line originating from *Spodoptera frugiperda*) and two mammalian cell lines (NIH 3T3 mouse fibroblast cells and HEK293 human kidney embryonic cells). Sf21 cells were purchased from Thermo Fisher Scientific (Riverside, CA, USA) and cultured in Sf-900 II SFM (Serum-Free Medium) (GIBCO, insect culture media from Invitrogen), supplemented with Antibiotic Antimycotic Solution (100 U mL$^{-1}$ penicillin, 100 µg mL$^{-1}$ streptomycin, 0.25 µg mL$^{-1}$ amphotericin B; Sigma) at 27 °C. NIH 3T3 cells (brand name "CRL-1658") were obtained from the American Type Culture Collection (ATCC; Manassas, VA, USA) and cultured in high glucose DMEM medium (Life Technologies), supplemented with 10% fetal bovine serum (FBS) and 1% penicillin/streptomycin at 37 °C. HEK293 cells (brand name "Freestyle HEK 293-F cells") were purchased from Thermo Fisher Scientific (Lyon, France) and cultured in FreeStyle$^{TM}$ 293 medium (Gibco) at 37 °C with 5% $CO_2$.

**Cytotoxicity assays**. Cytotoxicity of WT Cyt1Aa and of its mutants was assayed on HEK293 cells (Figs. 3 and 4e–g and Supplementary Fig. 14). Being mono-disperse in solution, these are indeed perfectly suited for cell toxicity assays at different concentrations using FACS flow cytometer systems. For Cyt1Aa WT, four conditions were tested: (1). Crystals solubilized in 0.1 M $Na_2CO_3$ buffer, pH 11.8, yielding a disulfide-linked dimer of Cyt1Aa; (2). Same as condition (1) supplemented with 1 mM DTT, yielding the 27.3 kDa (unactivated) protoxin monomer; (3). Same as condition (1) with activation by proteinase K for 1 h at 37 °C, yielding the 23 kDa activated toxin monomer; (4). Same as condition (3) supplemented with 1 mM DTT, to assess the impact of DTT on the 23 kDa activated toxin monomer. The same four conditions were tested for the mutants C7S and C190V. For all other mutants, only two conditions were tested: crystals solubilized in presence of DTT (monomeric protoxins) and toxins activated by proteinase K. A total of 10 doses was tested in each condition. For each dose of each WT or mutant toxin in each condition, a total of 1 million cells suspended in 1 mL of FreeStyle$^{TM}$ 293 medium (Gibco) were incubated with the toxin at different concentrations and with 1 µg mL$^{-1}$ of PI for 15 min. Cells were then inoculated into a MACS Quant VYB FACS (Milenyi Biotec) at a flow rate of 1000 cells s$^{-1}$. They were sorted according to their fluorescence at 617 nm (channel B2—built-in filter 589–639 nm range) upon laser excitation at 488 nm. Raw data were treated and extracted using the MACSQuantify software v2.11. Cell mortality was calculated by dividing the number of cells counted positive for PI staining by the total number of sorted cells (Supplementary Fig. 14). Mortality data were corrected using Abbott's formula to account for natural mortality in the control[56]. The best fitted model was selected among four logistic regression models for binomial distribution (logit, probit, complementary log-log transformation (cloglog) and Cauchy distribution) by comparing their deviance using a script modified from[54]. For each WT or mutant toxin species in presence or absence of DTT, $LC_{50}$ (doses leading to death of 50% of cell population) and 95% confidence intervals (CI) were calculated with the binomial distribution and a Pearson's chi square goodness-of-fit test, respectively, as implemented in the software R 3.5.2[42,54]. Differences in $LC_{50}$ between conditions were considered significant when 95% CI did not overlap[55].

**FITC-dextran cells exposure**. The adherent insect Sf21 and NIH 3T3 cell lines were used to visualize the Cyt1Aa-induced morphological changes, determine the dimension of membrane lesions and assess the specificity of Cyt1Aa mode of action (Fig. 5b, c and Supplementary Fig. 13). Sf21 and NIH cells were exposed to increasing concentrations of activated Cyt1Aa toxin, during 30 min at 27 °C and 37 °C respectively. Cells were then incubated with a solution of fluorescein isothiocyanate-dextran (FITC-dextran, or FD/from Sigma) at 4.5 mg.mL$^{-1}$. We selected two different dextran particles radius (FD-4, 1.4 nm and FD-150S, 8.5 nm). Cells were imaged using a laser scanning confocal microscope (LSM), Zeiss LSM 510, using an argon laser at 488 nm for the FITC-dextran solution (Laboratory for Fluorescence Dynamics, University of California Irvine, Irvine, CA). The emission was collected using a 500–550 nm band pass filter. For the time series, a solution of Cyt1Aa toxin plus FITC-dextran was incubated with the cells. The images were taken during the first 20 min of intoxication. A time series stack of a 100 consecutively scanned frames was collected at 1.2 s$^{-1}$. Once the stack was saved, a new series was collected—and so on, for an entire runtime of 20 min.

**BLM experiments**. The porous nature and permeability characteristics of Cyt1Aa MBO were tested by the BLM electrophysiological approach (Fig. 5a). Potassium Chloride (KCl) and N-cyclohexyl-3-aminopropanesulfonic acid (CAPS) were purchased from Sigma Aldrich Chemical Co. Inc. The multi-channel recording apparatus consisted of a two compartment Teflon chamber (~5 mL each) separated by a Teflon compartment with 300 µm diameter aperture for membrane formation. Bilayer lipid membranes were formed with 1% Asolectin in 4% Butanol in *n*-Decane. The aperture was pretreated with 2% Asolectin in chloroform and was allowed to cure for ~20 min to achieve solvent evaporation. The *trans*- and *cis*-sides

of the chambers were filled with buffer solution, 300 mM KCl, 10 mM CAPS at pH 9. Half a microliter of lipid (1% Asolectin in 4% Butanol in n-Decane) was added to the loop yielding a bilayer. Multi-channel reconstitution was achieved by addition of ~1 μL purified Cyt1Aa from a series of dilutions (final Cyt1Aa concentrations ranging from 1.7 to 60 μg mL$^{-1}$) into the *cis*-side of the chamber. Channel current traces were recorded with Ag/AgCl pellet electrodes (World Precision Instruments) with the *cis*-side of the chamber being the virtual ground, using the Axopatch 200B (Molecular Devices, LLC) patch-clamp amplifier in V-clamp mode (whole cell β = 1) with a CV-203BU headstage. The output signal was filtered by a lowpass Bessel filter at 10 kHz, and saved at a sampling frequency of 50 kHz using an Axon Digidata 1440A digitizer (Molecular Devices, LLC). Data analysis was performed with clampfit (Molecular Devices (USA).

**Oligomer formation and stoichiometry characterization**. To test the different conditions necessary for obtaining oligomers of Cyt1Aa protoxin and toxin, liposomes were prepared by the standard film-hydration method as previously described[57]. Briefly, liposomes were produced by drying L-α-phosphatidylcholine (POPC; Avanti Polar Lipids, France) under nitrogen flow to obtain a thin lipid film. Residual chloroform was eliminated by overnight vacuum. The lipid film was resuspended in phosphate-NaCl buffer (0.1 M K$_2$HPO$_4$/KH$_2$PO$_4$, 0.15 M NaCl, pH 7.4) by vortexing for 5 min. The multilamellar vesicles obtained were freeze-thawed (100–310 K) 20 times to obtain LUVs. Size calibration was performed either by sonication in an ice-cold water bath for 5 min (yielding ~50 nm radius small unilamellar vesicles or SUVs) or by extrusion using a mini-extruder (Avanti Polar Lipids, France) with 200 nm polycarbonate filters (yielding ~100 nm radius large unilamellar vesicles or LUVs). Homogenous size distribution was verified by dynamic light scattering (DLS) on a Wyatt DynaPro NanoStar. Prepared liposomes were directly used or stored for a maximum of 4 days at 4 °C until use.

The capacity of the three different forms (disulfide-bridged dimer, protoxin monomer and toxin monomer) of WT and mutants Cyt1Aa to generate oligomers upon contact with membranes was determined using SDS-PAGE 6% (Fig. 4c). Samples were mixed with Laemmli buffer devoid of DTT, electrophoresed for 90 min at 140 V, and revealed using InstantBlue (Sigma Aldrich, France). The three WT Cyt1Aa species that possibly co-exist in the mosquito larvae gut were tested, namely the disulfide-bridged protoxin dimer (Fig. 4c, lanes 1–2), the protoxin monomer (lane 3) and the activated toxin monomer (lanes 4–10). These were subjected to different treatments including addition of DTT at 10 mM (lanes 3 and 6); exposure to LUV (lanes 2–5, 5–7, 9 and 10) or SUV (lane 8); heating to 95 °C (lane 7); treatment with LDAO detergent at 2 times its CMC (Critical Micelle Concentration) before (lane 9) or after the addition of toxin to the LUV (lane 10). Kinetics of oligomerization were assayed by exposing the protoxin and toxin monomers to LUVs for 15 different durations (2, 5, 7, 10, 12, 15, 30, 45 min and 1 h, 1 h 30 min, 2 h, 2 h 30 min, 3 h, 3 h 30 min, 4 h) prior to loading on SDS-PAGE 6% (Supplementary Fig. 10). Gels were stained as described above and digitalized using a ChemiDoc XRS+ imaging system controlled by the Image Lab software version 6.0.0 (BioRad, France). The distance between each band in the oligomer population was determined using the software ImageJ v1.51k[41]. The influence of different detergents was investigated (Supplementary Fig. 11) by incubating pre-formed oligomers as described above with non-ionic (Tween-20, DDM, LDAO), zwitterionic (CHAPS, CHAPSO) and ionic detergents (SDS) at 2 times their CMC for 1 h. Samples were thereafter loaded on native PAGE (6% acrylamide), electrophoresed for 2 h at 110 V, and stained and digitalized as described above.

To determine the toxin unit form that underlies the formation of the oligomers (Fig. 4d), crystals of Cyt1Aa WT were solubilized in 0.1 M Na$_2$CO$_3$ buffer, pH 11.8. Protoxin monomers were generated by addition of 10 mM DTT, while activated toxin monomers were obtained by addition of trypsin at a final concentration of 0.1 mg mL$^{-1}$. Oligomers were formed upon addition of LUV (100 nm radius; 5 mg mL$^{-1}$ POPC) to the protoxin and toxin monomers, and incubation for 2 h at RT. The oligomer/LUV suspension was treated with LDAO at 4 mM final concentration (i.e., ~2 times its CMC in water) to disrupt all liposomes still intact. Solubilized oligomers were concentrated and purified using a 50 kDa cutoff AMICON ultra-filtration unit (Sigma Aldrich, France) which allowed to discard the remaining monomeric toxin, free POPC and LDAO monomers and mixed lipid/LDAO micelles. Oligomers formed either by the protoxin or the activated toxin, in the presence or the absence of DTT, were electrophoresed on a 15% SDS-PAGE gel either directly after incubation or after heating to 95 °C for 5 min thereby destabilizing oligomers, respectively. Unheated samples were also investigated by TLC enabling quantification of their lipid content (see below). Gels were stained and digitalized as described above. The relative intensities of bands corresponding to oligomer populations and to the monomer were calculated using the software ImageJ v1.51k[41]. To confirm SDS-PAGE results on unit basis of oligomers, unheated samples from condition 1 were also examined by MALDI-ToF using methods described above (Supplementary Fig. 6).

**Determination of oligomer lipid content by TLC**. TLC (Fig. 4d) was performed using 10 × 20 cm aluminum pre-coated TLC-sheets ALUGRAM Xtra SILGUR/ UV$_{254}$ (Macherey Nagel, France) containing a 0.2 mm layer of silica gel. Oligomer samples recovered after concentration and purification on a 50 kDa cutoff AMI-CON ultra-filtration unit (see above) were loaded on the concentrating zone (layer of kieselgur) of the TLC sheets. Two concentrations and two duplicates were tested

for each sample and a concentration range of lipids (from 0.5 to 50 μg) was also deposited for lipid concentration determination. Samples were allowed to migrate in a glass tank containing 50 mL of a 65:35:5 chloroform:methanol:water mixture. After 40 min. migration, phospholipids were specifically stained by spraying molybdatophosphoric acid (1–5% in ethanol; Macherey-Nagel, France) on the sheets and revealed by heating using a heat block. The intensity of each band was analyzed using the software ImageJ v1.51k[41].

**Oligomer visualization by TEM**. For negative stain TEM observations (Fig. 5n–q), 3 μL of sample were applied to the clean side of carbon on a carbon-mica interface and stained with 2% sodium silicotungstate. Micrographs were recorded on a FEI Tecnai T12 microscope operated at 120 kV with a Gatan Orius 1000 camera, at a nominal magnification of ×49,000 resulting in a pixel size of 1.26 Å. For cryo-EM observations, 3 μL of sample were applied to glow-discharged quantifoil grids 300 mesh 1.2/1.3 (Quantifoil Micro Tools GmbH, Germany), excess solution was blotted with a Vitrobot (FEI) and the grid frozen in liquid ethane[58]. Data collection was performed on a FEI F20 microscope operated at 300 kV under low dose conditions. Images were recorded on a CETA camera (FEI) at a nominal magnification of ×50,000 corresponding to a pixel size of 2.09 Å. Images were converted into the TIFF format and then imported into the software ImageJ v1.51k[41] to calculate the length and curvature of oligomers.

**Kinetics of toxin oligomerization and membrane perforation**. Interaction between Cyt1Aa activated toxin and reconstituted SLB was imaged on the same Multimode 8 AFM as described above in fluid conditions (Fig. 5d–m). On a freshly cleaved mica, a drop of 36 μL of buffer (100 mM Phosphate, 150 mM KCl) was deposited followed by a drop of 4 μL of previously prepared liposomes. This preparation was left at room temperature for 2 h or overnight in a humid environment. Weakly bound material was removed by rinsing three times with 40 μL of buffer (at each rinse, 40 μL is removed and subsequently 40 μL of fresh buffer is added). Before adding the protein, a drop of 40 μL of fresh buffer was deposited on the supported bilayers on the mica. Native membranes were imaged first and if a proper fusion was observed (large areas of uninterrupted supported bilayers), the drop was replaced by 40 μL of fresh buffer and 4 μL of the toxin at a concentration of 4 mg mL$^{-1}$ was injected in the drop. ScanAsyst fluid cantilevers ($k = 0.7$ N m$^{-1}$, Fq = 150 kHz, nominal tip radius = 20 nm, Bruker probes, Camarillo, CA, USA) were used. 512 × 512 pixel images were collected at ~1 Hz rate, with scan sizes varying from 7 to 1.7 μm in length, using the ScanAsyst mode in a semi-automatic condition where both the gain and the set-point (from 40 to 120 mV) were manually adjusted. The default ramp size for the peak-force mode was kept at 150 nm.

TIFF images were imported into the software ImageJ v1.51k[41] to calculate the surface of each oligomer as a function of elapsed time since toxin addition. Considering that the data did not follow a normal distribution (Shapiro–Wilk test, $W = 0.92$, $p < 2.2 \times 10^{-16}$, $W = 0.96$, $p = 0.12$; $W = 0.91$, $p < 2.2 \times 10^{-16}$), the linear correlation between (i) the size of the oligomer and the elapsed time since toxin addition (Fig. 5g); (ii) the hole depth and the hole surface (Fig. 5j); and (iii) the hole depth and the hole + MBA surface (Fig. 5k) was tested using non-parametric Spearman's Rho rank correlation coefficients, as implemented in the software R 3.5.2[42]. Plot correlograms and 95% confidence regions were generated using the "ggpubr", "corrplot" and "Hmisc" libraries.

**Reporting summary**. Further information on research design is available in the Nature Research Reporting Summary linked to this article.

## Data availability

Structures and structure factor amplitudes have been deposited in the PDB databank under accession codes 6T14 ("pH7"; [10.2210/pdb6T14/pdb]), 6T19 ("DTT"; [10.2210/ pdb6T19/pdb]), 6T1A ("pH10"; [10.2210/pdb6T1A/pdb]) and 6T1C ("C7S mutant"; [10.2210/pdb6T1C/pdb]). The source data for Figs. 3, 4a–g, 5d–h and 5j–q and for Supplementary Figs. 5, 7–12 and 14b are provided as a Source Data file. Other data are available from the corresponding author upon reasonable request.

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

## Acknowledgements
The authors thank Rida Awad (IBS[1]) for generously providing the HEK293 cells used in toxicity assays; Jean-Philippe Kleman (IBS[1]) for his support in using the FACS system and performing raw data analysis; Ilme Schlichting (Max-Planck Institute, Heidelberg), David S. Eisenberg (UCLA[10]), and Michel Vivaudou (IBS[1]) for critically reading the manuscript; and Raymond Barrett, Amparo Vivo (ESRF[4]) and Lahsen Assoufid (Argone National Laboratory) for providing the KB-mirror that was used in the serial synchrotron

micro-crystallography experiment (Supplementary Fig. 1) and for assistance with its installation at ID13. IBS[1] acknowledges integration into the Interdisciplinary Research Institute of Grenoble (IRIG, CEA). This work was supported by the Agence Nationale de la Recherche (grants ANR-17-CE11-0018-01 and ANR-2018-CE11-0005-02 to J.-P.C.), the CNRS (PEPS SASLELX grant to MW) and used the AFM platform at the IBS and the platforms of the Grenoble Instruct-ERIC center (ISBG; UMS 3518 CNRS-CEA-UGA-EMBL) within the Grenoble Partnership for Structural Biology (PSB). Platform access was supported by FRISBI (ANR-10-INBS-05-02) and GRAL, a project of the University Grenoble Alpes graduate school (Ecoles Universitaires de Recherche) CBH-EUR-GS (ANR-17-EURE-0003). B.F. acknowledges funding support from the Pacific Southwest Regional Center of Excellence for Vector-Borne Diseases funded by the U.S. Centers for Disease Control and Prevention (Cooperative Agreement 1U01CK000516) and U.S. National Institutes of Health grants UO1 AI054778 and RO1 AI045817. This research was also supported by NIH grant GM117126 to N.K.S. We thank the LCLS for beamtime allocation under proposals P125 and P141. Use of the LCLS at SLAC National Accelerator Laboratory, is supported by the US Department of Energy, Office of Science, and Office of Basic Energy Sciences under contract no. DE-AC02-76SF00515. The CXI instrument was funded by the Linac Coherent Light Source Ultrafast Science Instruments project, itself funded by the DOE Office of Basic Energy Sciences. Parts of the sample injector used at LCLS for this research were funded by the National Institute of Health, P41GM103393, formerly P41RR001209.

## Author contributions

J.-P.C designed and coordinated the project; G.T., A.-S.B., E.A.A., N.B., H.W.P. and B.F. designed and constructed plasmids; G.T., A.-S.B., E.A.A., N.B., F.L. and L.D. produced Cyt1Aa crystals in vivo; I.S. performed crystal visualization by SEM; D.F., J.-M.T., and J.-L. P. performed crystal visualization by AFM; A.-S.B. and J.-P.C. secured beamtime at the ESRF; G.T., A.-S.B., E.A.A., T.G., M.R., M.Bu., and J.-P.C. performed serial data collection at the ESRF; G.T., A.-S.B., E.A.A., D.C., M.W., M.S., and J.-P.C. secured beamtime at the LCLS; G.T., A.-S.B., E.A.A., A.S.B., R.G.S., I.D.Y., S.B., D.C., M.R.S., N.K.S., M.S.H., and J.-P.C. performed serial data collection at the LCLS; J.-P.C. performed atomic model building, refinement and structure interpretation; A.S.B., I.D.Y., T.G., N.C., M.Bu., N.K.S. produced new processing tools or devices; A.S.B., I.D.Y., N.K.S., and J.-P.C. performed serial data processing; R.G.S. developed and operated the MESH-on-a-stick injector; L.S. and E.B.E. performed mass spectrometry experiments; G.T., A.-S.B., and E.A.A. performed solubilization assays; G.T. and E.A.A. performed heat stability assays; G.T., J.B., MT.F-L., M.D., E.G., R.S., N.Z., and B.F. performed cytotoxicity assays; A.B., M.Ba., and I.G. conducted transmission electron microscopy imaging; J.-M.T. and J.-L.P. conducted atomic force microscopy imaging on membranes; G.T., J.A.B., and M.W. conducted black lipid membrane experiments; G.T. performed the statistical analyses; G.T. and J.-P.C. prepared the manuscript with input from coauthors.

## Competing interests

The authors declare no competing interests.
