## [Peer Review File · Nature Communications]

Reviewers' comments:

Reviewer #1 (Remarks to the Author):

This manuscript is a massive body of work that starts with the excellent description of the protoxin form of Cyt1Aa, an important anti-mosquito pore forming protein. This builds on previous publications of the activated form of Cyt1Aa which lacks the pro-peptide in the crystal structure. The novelty of this manuscript also comes in the form of a high level of well-thought out experiments that probe the role of pH and redox environment in the dissolution of the in vivo crystals in the pathway prior to proteolytic activation. The final part of the manuscript presents a model for pore formation. Overall there is significant progress presented in understanding how the Cyt1Aa protein goes from being produced within the bacteria as crystals to the active monomeric form. There is some effort to then characterise the formation of oligomers and membrane-embedded pores.

The high standard of experimentation should also be a good influence on the wider structural biology field where that study protein activation and complex formation. This is a novel piece of work firstly in the structure of the pro-toxin form. But secondly in the high level of interrogation of the structure leading to well-designed experiments that map each step of the pathway from in vivo crystals to formation of active monomers. Overall the data is very convincing. The appropriate conclusions were drawn for how the crystal dissolves and turns into active monomeric protein. However, the final discussion presents a model that is too speculative and not supported at all by the data shown in the manuscript.

MAJOR COMMENTS

Lines 383 – 397: This ends on a proposed molecular model is far too speculative and it the kind of thing that leads to erroneous dogma in the field. This needs to be left out. If authors want to pursue this kind of model then they would need either high resolution structures of the pore or use lipid sensitive dyes. For examples of this kind of research see how it was done for CDCs (The mechanism of membrane insertion for a cholesterol-dependent cytolysin: a novel paradigm for pore-forming toxins. Shatursky O, Heuck AP, Shepard LA, Rossjohn J, Parker MW, Johnson AE, Tweten RK. *Cell*. 1999 Oct 29;99(3):293-9 AND Shepard LA, Heuck AP, Hamman BD, Rossjohn J, Parker MW, Ryan KR, Johnson AE, Tweten RK. *Biochemistry*. 1998 Oct 13;37(41):14563-74). The use of disulphide trap mutants would also be an indirect method of investigating which bits insert into the membrane.

The word/phrase life cycle implies a pathway that goes back to the start, e.g. for organisms that start as infants then become adults then reproduce to make more infants. For the title and elsewhere in the paper, it would be better to use alternative phrases. E.g. Life span or assembly pathway. I am happy to make more suggestions.

In the abstract and introduction, the authors allude to the "new model reconciling previously opposed data"(line 44 and lines 69-70). This manuscript would be significantly stronger if the conflicting models were presented and referenced. I can see this is hinted at in lines 381-2 in the discussion but needs to be introduced as a concept in the introduction rather than being fresh material in the Discussion section. It could also be considered for the abstract. This is important to show the research community how very careful experimentation is important in preventing and disproving false lines of research. This is particularly important for the wider class of pore forming proteins due to the complexity of the assembly pathways.

The abstract could also be more specific in the forms of protein characterisation rather than stating "Complementary characterisation". This is an unnecessarily vague comment for an abstract.

The multitude of interfaces within the crystal structure are difficult to follow and need to be well

supported with the main figures and supplementary figures. Where possible, make the figures describing the electron density and the H-bonds larger, especially figure 2. It would be very helpful for the reader who is unfamiliar with the fold of the protein to label the discussed secondary structure on the central orange and blue structures in Fig 2a. e.g. where is alphaC/alphaD? The insert box does not seem to correspond to the same region for this part of the figure.

Figure 13, the movement of alphaC/alphaD region between the pro-toxin and toxin structures is not obvious in the figure.

The description of the fold is incorrect in that it is not a true beta-sandwich, such as seen for 3 domain Cry toxins, aegerolysins and the domain 4 of CDCs, which all have two beta-sheets in a face-to-face orientation). Instead it would be better described as having a α/β architecture with one beta-sheet. This β -sheet is surrounded on both faces with α -helices. (I found a decent description in the following review: *Toxins* 2014, 6(9), 2732-2770; "Structural Insights into *Bacillus thuringiensis* Cry, Cyt and Parasporin Toxins" but you could consider using CATH for best possible description of the fold.)

Can an explanation be provided for why the pH 10 condition becomes mildly reducing in order for the disulphide bond to break? What published evidence of this happening in alkaline conditions?

The relative change in pH sensitivity across the different mutants is great work, however, it would be best to avoid definitive statements about if a crystal will dissolve in certain conditions, e.g. results section (lines 161-162) and conclusions (line 349-51). The pH dissolution experiments presented in this manuscript whether a crystal would dissolve at certain pH environments were made based on a set time point of 1 hour (and at the given temperature was not provided). It cannot be ruled out that lower pH values cannot completely dissolve the crystal if given longer periods of time. Then again, how long does it take for food to pass through the mosquito larvae digestive tract and does the whole crystal need to be dissolved to be effective?

Line 372-3. The referenced publication used as evidence is 20 years old and I would consider it as rather speculative. Are there more recent independent publications of beta-sheet insertion from beta-strands 5, 6 and 7? Referencing of this paper seems to contradict the very speculative section of the conformational change (roughly lines 385 – 396, see next paragraph)

MINOR POINTS

Line 76: "allows formulating a" -> " allows formulation of a"

Line 69-70 has poor sentence construction and grammar. Replace "yet remain as obfuscated as is actively debated" could be replaced with "yet remain unclear as it is actively debated"?

Line 101: beta-3 instead of beta-4?

Curiosity question: What is the % solvent of the activated toxin crystal (leading to the 3RON structure) compared with the in vivo pro-toxin crystal form presented in this manuscript? From a crystallographer's perspective it would be interesting to see how packed the protein is in the crystal. Especially since they are so resistant to dissolution compared to the average protein crystal.

Figure 3b: typo "dilsulfide"

Supp Fig 7; please use different colours or shapes instead of shades of the same colour

Lines 307 and 308 refer to Fig 4 not Fig 5

Line 315: deepness -> depth

Methods: Make sure there are units are numbers in methods and that the tense is consistent (i.e. not switching between past and present tense)

Line 477: Is this a percent or a ratio?

There are essentially results presented in the methods section on pages 21-23. Should this be moved to the supplementary section as further analysis?

Line 801 "Not strongly bound material" -> "Weakly bound material"

Reviewer #2 (Remarks to the Author):

The AFM data presented in this work is good and robust.

The measurements are well performed and the interpretation seems to be correct. In particular, the following results are interesting for researches working on the interactions between toxin proteins and lipid bilayers: "the structural transition between MBA and porous oligomers is independent from MBA size and could involve formation of a pre-pore", and "the Q168E mutant is unable to penetrate the membrane and aggregate to form pores"

I would recommend the paper for publication.

I only have two minor comments:

1. The toxin adsorption kinetics and the final toxin-bilayer structure could depend on the initial toxin concentration. Please, indicate in the text (or in the figure caption) the used toxin concentration.

2. If possible, show force-distance curves to demonstrate the bilayer existence (which should match with the profile of figure 4i). Also, other force-distance curves could deliver information about structural changes in the toxin-bilayer with time.

Reviewer #3 (Remarks to the Author):

This manuscript described the fine-structure of Bti natural parasporal crystal consisting of Cyt1Aa protein at the nano-level by using serial femtosecond nano-crystallography related techniques. Further more, this ms described the structure transition and dynamic of the protoxin, activated toxin and member-bound oligomer, providing many more new and fine knowledge about the structure of parasporal crystal, the processing pathway of protoxin, and the pore-formation.

1. We can know about the native structure of the crystal formed by Bt bacterium. Previously, we just can know the structure of crystal protein in detail. But we can not image the native structure after the crystal protein forms granular crystal. This work showed Cyt1Aa protoxin crystallizes as a domain-swapped dimer with strands $\beta 1$ and $\beta 2$ at the DS interface #1. Even we do not know why this dimer accumulates a bypyramidal shaped crystal.

2. This work described the related concern about the dissolution of crystal and the processing path, which concerning the alpha and beta domain, H-bond, disulfide bridge, N-terminal part as well as pH value and reducing power.

3. This work described how toxin accumulate on the member surface, aggregate as an oligomer and form porous conformer with 54nm diameter by 56 mono proteins.

4. This work use Sf21 insect cells and mammalian cells instead of mosquito midgut cells to elucidate the binding\aggregation\pore formation, and explain why Cyt1A serves as a receptor for 3D domain crystal protein Cry4Ba or Cry11Aa to insertion in membrane (line 406-407). If it is right, Cyt1A could also serve as a receptor for 3D domain crystal protein Cry1Aa to insertion in membrane and synergizes Cry1Aa toxin by functioning as a membrane-bound receptor. It is valuable to test this. It is unknown that the Sf cells is same as the mosquito cell when Cyt1A toxin play function on membrane. Even Cyt1A toxin show cytotoxic activity, it only exists in Bti strains which are highly toxic to diptera insects. There is not any Cyt1A toxin existing in Btk strains, which contain 3D domain Cry1A crystal proteins and are highly toxic to lepidoptera insects. If Btk strain

obtained cyt1A gene, it would help Btk overcome insect resistance.

I highly value this work. It will push the work to elucidate the fine structure of all Bt naturally-crystal especially the highly toxic lepidopteran-specific Cry1A toxins, and further to know how and why the crystal protein can naturally form crystalline granule. The crystal protein accounts for 20-30% of the all proteins of dry weight of sporulation culture. If we can let the proteins we need to form crystal inside Bt strain upon the mechanism of crystal formation, we will set up a new expression system as crystal style with Bt system comparing to E.coli system. The crystal structure base on the processing path of crystal in insect gut and the manner of toxin binding, aggerating and perforating will facilitate to know the more detail action model and to help to setup the insect resistance management.

Minor concern:

This work described the natural crystal as a nano-crystal. In my knowledge, nano scale means those less than 100nm. While the size of Bt natural crystal often larger than 100nm and near to 1000 nm. In this work, the Cyt1Aa natural crystal is 925x591 nm as described in table 2.

Reviewer #1

This manuscript is a massive body of work that starts with the excellent description of the protoxin form of Cyt1Aa, an important anti-mosquito pore forming protein. This builds on previous publications of the activated form of Cyt1Aa which lacks the pro-peptide in the crystal structure. The novelty of this manuscript also comes in the form of a high level of well-thought out experiments that probe the role of pH and redox environment in the dissolution of the *in vivo* crystals in the pathway prior to proteolytic activation. The final part of the manuscript presents a model for pore formation. Overall there is significant progress presented in understanding how the Cyt1Aa protein goes from being produced within the bacteria as crystals to the active monomeric form. There is some effort to then characterise the formation of oligomers and membrane-embedded pores.

The high standard of experimentation should also be a good influence on the wider structural biology field where that study protein activation and complex formation. This is a novel piece of work firstly in the structure of the pro-toxin form. But secondly in the high level of interrogation of the structure leading to well-designed experiments that map each step of the pathway from *in vivo* crystals to formation of active monomers. Overall the data is very convincing. The appropriate conclusions were drawn for how the crystal dissolves and turns into active monomeric protein. However, the final discussion presents a model that is too speculative and not supported at all by the data shown in the manuscript.

Response. We warmly thank the referee for his/her comprehensive and positive review. All minor and major concerns have been addressed. All modifications are highlighted in the revised manuscript. Below, we offer a point-by-point response to his/her comments and queries.

MAJOR COMMENTS

Lines 383 – 397: This ends on a proposed molecular model is far too speculative and it the kind of thing that leads to erroneous dogma in the field. This needs to be left out. If authors want to pursue this kind of model then they would need either high resolution structures of the pore or use lipid sensitive dyes. For examples of this kind of research see how it was done for CDCs (The mechanism of membrane insertion for a cholesterol-dependent cytolysin: a novel paradigm for pore-forming toxins. Shatursky O, Heuck AP, Shepard LA, Rossjohn J, Parker MW, Johnson AE, Tweten RK. *Cell*. 1999 Oct 29;99(3):293-9 AND Shepard LA, Heuck AP, Hamman BD, Rossjohn J, Parker MW, Ryan KR, Johnson AE, Tweten RK. *Biochemistry*. 1998 Oct 13;37(41):14563-74). The use of disulphide trap mutants would also be an indirect method of investigating which bits insert into the membrane.

Response. We agree with the referee that there is a thin line between interpretation of data and speculation. We also believe it is the role of structural biologists to interpret their observations beyond an atomistic description and propose molecular mechanisms that are consistent with previous knowledge and the novel data presented in this work. We agree with the referee that the line was short to be crossed in our initial manuscript when we proposed that “the MBA conformer would have the α C/ α D region and the hydrophobic side of the β -sheet laying on the membrane surface, while the α A/ α B helices and side of the β -sheet would be

exposed to the bulk solvent” and that “residues in the α D- β 4 loop [would] adopt an extended conformation (β^*) to complement β 4 in forming a strand long enough to lay aside β 5, β 6 and β 7 in four-stranded β -sheet ($\beta^*\beta$ 4- β 5- β 6- β 7) now capable of fully spanning the membrane”. Accordingly, we have removed these lines from the Discussion. We also have removed Supplementary Fig. 15, which offered a picturing of this proposed MBA-to-porous oligomer transition, from our revised version of the manuscript.

However other points in our Discussion, based on strong published data from other (widely-esteemed) colleagues in the field cannot be deemed speculative. Briefly, Ellar and coll. demonstrated that residues 154-234, corresponding to the α D- β 4 loop and the β 4- β 5- β 6- β 7 β -sheet featuring the short α E helix between β 6 and β 7, are membrane inserted and inaccessible to proteases [Du et al., 1999, Biochem J], and that the N- and C-termini of the protein are both exposed on the extracellular side of the membrane [Promdonkoy & Ellar, 2005, Mol Membr Biol]. These results establish that : (i) the α D- β 4 loop and β 4 are involved in the core structure of oligomeric pore; (ii) there is an even number of strands in the membrane-spanning β -sheet of the Cyt1Aa porous conformer and the β 4- β 5 loop and α E helix sit on the intracellular side of the pore, since both the soluble N-terminal β 2- α A- α B- β 3 segment and the C-terminal α F loop are found on the extracellular side of the pore; and (iii) β 2 and β 3 must at some point dissociate from β 5 and β 7 at the two sides of the central β -sheet, respectively, leaving β 4- β 5- β 6- β 7 inserted in the membrane. Our model builds on these results and conclusions, which allow to map the membrane insertion topology of Cyt1Aa even in the absence of a high resolution (cryoEM) structure.

We believe that we sowed confusion by discussing these literature results as supporting points for the presentation of our model, when we could/should have discussed them separately. This is accordingly what we do in the revised version of our manuscript. Briefly, we extend our third paragraph of the Discussion, dealing with results obtained on the Q168E mutants and their implications for the role of the β 4- β 5 loop in membrane perforation, by a more complete presentation of the above-mentioned results by Ellar and coll. and others. We separate the presentation of these literature results and their interpretation of the presentation of our model, summarized by Fig. 5, which summarizes all the steps for which either we present definitive experimental support and/or a consensus has emerged in the literature and wherein we hardly emit molecular level hypothesis. Our only remaining ‘speculation’ is the proposal that the α D- β 4 loop may in some way assist β 4 in forming a longer β -strand, thereby helping to stabilize the pore oligomeric architecture by perfecting the junction between adjacent β -sheet. We also acknowledge the limits of our data, and the need for a high resolution (cryoEM) structure of the pore. We believe that our revision makes a clear distinction between results and

discussion, and between discussion and speculation. We are grateful to the referee for his/her help in attaining this.

The word/phrase life cycle implies a pathway that goes back to the start, e.g. for organisms that start as infants then become adults then reproduce to make more infants. For the title and elsewhere in the paper, it would be better to use alternative phrases. E.g. Life span or assembly pathway. I am happy to make more suggestions.

Response. We initially used this term based on the Cambridge Dictionary defining life cycle as “*the series of changes that a living thing goes through from the beginning of its life until death*”. However, we agree with the referee on the possible confusion that the use of the term “life-cycle” sows given that proteins, including Cyt1Aa, cannot be considered as “living thing”. Therefore, we have replaced the term “life-cycle” by “bioactivation cascade”, throughout the manuscript. Accordingly, our title now reads “Serial femtosecond crystallography drives elucidation of mosquitocidal Cyt1Aa bioactivation cascade, from *in vivo* crystallization to cell lysis.”

In the abstract and introduction, the authors allude to the “new model reconciling previously opposed data”(line 44 and lines 69-70). This manuscript would be significantly stronger if the conflicting models were presented and referenced. I can see this is hinted at in lines 381-2 in the discussion but needs to be introduced as a concept in the introduction rather than being fresh material in the Discussion section. It could also be considered for the abstract. This is important to show the research community how very careful experimentation is important in preventing and disproving false lines of research. This is particularly important for the wider class of pore forming proteins due to the complexity of the assembly pathways.

Response. We thank the referee for this suggestion. We now present the two proposed models for pore formation by Cyt1Aa, at the end of the first paragraph of the Introduction. However, we refrained from introducing this controversy in the Abstract due to the limited word count and our need to highlight key experimental results. Hence, we also removed reference to our “new model reconciling previously opposed data” in the Abstract.

The abstract could also be more specific in the forms of protein characterisation rather than stating “Complementary characterisation”. This is an unnecessarily vague comment for an abstract.

Response. We thank the referee for this suggestion. We have replaced “Complementary characterization enables...” by “Biochemical, toxicological and biophysical methods enable...”. We considered the alternative of stating all used methods (e.g. “Electrophoresis, mass-spectrometry, cytotoxicology, electrophysiology, atomic-force microscopy and electron microscopy enable...”) but it was discarded on the basis that such a list would confuse readers rather than meliorate their understanding.

The multitude of interfaces within the crystal structure is difficult to follow and need to be well supported by the main figures and supplementary figures.

Response. We agree with the referee on the importance of describing the various crystal interfaces as precisely as possible. In line with this, we have withdrawn Supplementary Fig. 3b, c, where the interfaces were only briefly presented, and introduced a new Supplementary Table 1, where we now describe in detail and illustrate all crystal packing interfaces. This has allowed to make Supplementary Fig. 3a (now Supplementary Fig. 3) larger, offering a more detailed view to readers.

Where possible, make the figures describing the electron density and the H-bonds larger, especially figure 2.

Response. It is difficult to make the panels in Fig. 2 larger without jeopardizing the information content of the assembled figure. We nonetheless agree on the importance of showing the difference density observed upon pH elevation and soak with DTT. We have found that the best way to present this figure is with it occupying the full width of a printed page; should our manuscript be accepted, we would thus convene with the editor so as to have Fig. 2 occupy such a space. We also present these difference density data in more detail in the now-enlarged Supplementary Fig. 4.

It would be very helpful for the reader who is unfamiliar with the fold of the protein to label the discussed secondary structure on the central orange and blue structures in Fig 2a. e.g. where is alphaC/alphaD?

Response. The secondary structure information is presented in Fig. 1e. Adding it onto Fig. 2a, which aims at familiarizing readers with the disulfide-chaining of domain-swapped dimers, would probably be more confusing than informative. However, it was possible to add this information in the panels of Fig. 2b, where it is even more useful for readers.

The insert box does not seem to correspond to the same region for this part of the figure.

Response. This is a perspective effect; the insert box indeed corresponds to the same region for this part of the figure albeit after a 180° rotation along the horizontal direction (as stated in the figure).

Figure 13, the movement of alphaC/alphaD region between the pro-toxin and toxin structures is not obvious in the figure.

Response. We believe the referee is here referring to either Fig. 1e or Supplementary Fig. 2b, since this is where we compare the protoxin and activated toxin structures. We have added a panel in Supplementary Fig. 2 (new panel b), to highlight the subtle but concerted motions of

C α atoms of the protoxin upon activation. Accordingly, we added a sentence referring to this figure at the end of the first paragraph of the Results section, which reads: "While these conformational changes are on the overall subtle, they appear to be concerted at the main-chain level (Supplementary Fig. 2b)". We also now highlight side chain conformational changes of interest by red circles in Supplementary Fig. 2 (new panel c). We last added a panel in Supplementary Fig. 2 (new panel d) where we show the concerted motions of C α atoms in the protoxin upon pH elevation. In all three figures, we have included secondary structure information so as to familiarize readership with the structure of the protein.

The description of the fold is incorrect in that it is not a true beta-sandwich, such as seen for 3 domain Cry toxins, aegerolysins and the domain 4 of CDCs, which all have two beta-sheets in a face-to-face orientation). Instead it would be better described as having a α/β architecture with one beta-sheet. This β -sheet is surrounded on both faces with α -helices. (I found a decent description in the following review: *Toxins* 2014, 6(9), 2732-2770; "Structural Insights into *Bacillus thuringiensis* Cry, Cyt and Parasporin Toxins" but you could consider using CATH for best possible description of the fold.)

Response. We thank the referee for noting this point. We note that the fold is properly and extensively described in the first paragraph of the Result section. We nonetheless agree that the description of Cyt1Aa as a beta-sandwich is inaccurate and we accordingly have corrected the two occurrences where the term 'beta-sandwich' was used.

Can an explanation be provided for why the pH 10 condition becomes mildly reducing in order for the disulphide bond to break? What published evidence of this happening in alkaline conditions?

Response. It has indeed been shown that disulfide bonds can be ruptured by oxidation upon pH elevation, especially those that are solvent accessible, by the following reaction: Cys-S-S-Cys \rightarrow Cys-S- + Cys-S-OH (Setterdahl et al., 2003, JACS; Florence, 1980, Biochem J). Disulfide rupture at alkaline pH has notably been used to follow the unfolding kinetics of lysozyme (Ravi, Goel, Kotamarthi et al., 2014, Plos ONE; Kumar, Ravi, Swaminathan et al., 2003, Biochemistry). We now have added this information to the paper.

The relative change in pH sensitivity across the different mutants is great work, however, it would be best to avoid definitive statements about if a crystal will dissolve in certain conditions, e.g. results section (lines 161-162) and conclusions (line 349-51). The pH dissolution experiments presented in this manuscript whether a crystal would dissolve at certain pH environments were made based on a set time point of 1 hour (and at the given temperature was not provided). It cannot be ruled out that lower pH values cannot completely dissolve the crystal if given longer periods of time.

Response. We completely agree with the referee that sensitivity of crystals to pH is time-dependent. The SP50 reported in our manuscript are pH values at which 50% of crystal solubilize at RT (22°C) after one hour of incubation in buffers. This point is now clearly stated in the manuscript (Results and Methods section).

Then again, how long does it take for food to passage through the mosquito larvae digestive tract and does the whole crystal need to be dissolved to be effective?

Response. The transit time along the mosquito larvae gut is 30-60 minutes, depending on species (Walker, 1995, J Am Mosqu Contr Assoc). This explains our choice of estimating the SP50 by measuring the amount of toxin solubilized after one hour of incubation of crystals at various pH. We have now added this information to the paper. It was also shown that the parasporal envelope is emptied after evacuation from the mosquito hindgut (Diaz-Mendoza et al., 2012, J Bacteriol), suggesting that the whole crystal is dissolved during the transit of *Bti* crystals along mosquito larvae gut.

Line 372-3. The referenced publication used as evidence is 20 years old and I would consider it as rather speculative. Are there more recent independent publications of beta-sheet insertion from beta-strands 5, 6 and 7?

Response. We added two additional references supporting the *pore-forming* model at the end of the first paragraph of the Introduction and at the beginning of the fourth paragraph of the Discussion.

Referencing of this paper seems to contradict the very speculative section of the conformational change (roughly lines 385 – 396, see next paragraph)

Response. This comment echoes an earlier remark by the referee related to the speculative nature of the proposed model for the insertion of Cyt1Aa in the membrane. As stated above, we now have withdrawn the problematic sentences and figure (old Supplementary Fig. 15) and have reorganized the presentation of the discussions of previous results, on the one hand, and of our model, on the other hand, so that this comment is addressed as well.

MINOR POINTS

Line 76: “allows formulating a” -> “ allows formulation of a”

Response. We thank the referee for noting this point, which has been changed in the revised version of the manuscript.

Line 69-70 has poor sentence construction and grammar. Replace “yet remain as obfuscated as is actively debated” could be replaced with “yet remain unclear as it is actively debated”?

Response. The sentence has been rewritten for clarity, so that it now reads: “However, the molecular determinants of Cyt1Aa crystallization in *Bti* cells and of crystal dissolution in the mosquito midgut remain unclear, and the mechanism by which oligomers form and exert direct toxicity to mosquito gut cells is actively debated.”

Line 101: beta-3 instead of beta-4?

Response. We thank the referee for noting this typo which has been corrected in the revised manuscript.

Curiosity question: What is the % solvent of the activated toxin crystal (leading to the 3RON structure) compared with the *in vivo* pro-toxin crystal form presented in this manuscript? From a crystallographer’s perspective it would be interesting to see how packed the protein is in the crystal. Especially since they are so resistant to dissolution compared to the average protein crystal.

Response. Crystals of the protoxin (pdb entry 6t14) and activated toxin (pdb entry 3ron) feature roughly the same solvent content, viz. 34.1 and 31.7 % respectively, underlining the poor solubility of the toxin in the absence of its propeptide (see Cohen et al., 2011, Prot Science for a description of the precipitation-based strategy used to isolate the Cyt1Aa monomers that were then used to grow crystals *in vitro*) and that solvent content is a poor metric to explain the stability of *in vivo*-grown crystals. For example, *in vivo*-grown crystals of the 93 kDa binary toxin BinAB feature 60% solvent content, but are as stable as the Cyt1Aa protoxin crystals (i.e. up to pH 10.5). Most relevant to the differential stability of *in vivo*-grown (protoxin) and *in vitro*-grown (activated toxin) crystals is the per-monomer buried surface area at protein-protein interfaces in the crystals, which is 5484.2 Å² for the protoxin and 1252.7 Å² for the activated toxin, respectively corresponding to 40.1 and 13.3 % of the total monomeric area (the total areas per protoxin and activated toxin monomers are 13661.2 and 9405.5 Å², respectively). We believe that this information could also be of interest for the general readership and have therefore included a sentence specifying the per-monomer buried surface area in the protoxin crystals grown *in vivo* and the activated toxin crystals grown *in vitro*, at the beginning of the second paragraph of the Results section. Specifically, the sentence reads: “*In vivo*-grown Cyt1Aa protoxin crystals are remarkably packed, burying 40.1% of surface area at crystal contacts (5484.2 Å² out of 13661.2 Å² of total monomeric areas), compared with only 13.3% (1252.7 Å² out of 9405.5 Å²) in crystals of the activated toxin grown *in vitro*.”

Figure 3b: typo “dilsulfide”

Response. We thank the referee for noting this typo which has been fixed in the revised manuscript.

Supp Fig 7; please use different colours or shapes instead of shades of the same colour

Response. We now use a cold-to-warm color palette to present the heat-stability at different pH of the Cyt1Aa protoxin dimer. The caption of the figure has been modified accordingly.

Lines 307 and 308 refer to Fig 4 not Fig 5

Line 315: deepness -> depth

Response. We thank the referee for noting these typos which have been fixed in the revised manuscript.

Methods: Make sure there are units are numbers in methods and that the tense is consistent (i.e. not switching between past and present tense)

Response. The Material and Methods section is now presented in the past tense exclusively. We also verified that units are present for all numbers in this section (and throughout the manuscript).

Line 477: Is this a percent or a ratio?

Response. The value referred to by the referee is a mass per volume ratio (grams of crystals per 100 mL of buffer solution); this is now explicitly stated in the manuscript.

There are essentially results presented in the methods section on pages 21-23. Should this be could be moved to the supplementary section as further analysis?

Response. We thank the referee for this suggestion. Accordingly, we have moved our section entitled "A difference-density based mutation strategy" from the Material and Methods to the Supplementary material (new Supplementary Results section).

Line 801 "Not strongly bound material" -> "Weakly bound material"

Response. We thank the referee for this suggestion which was followed in our revision.

Reviewer**#2**

The AFM data presented in this work is good and robust.

The measurements are well performed and the interpretation seems to be correct. In particular, the following results are interesting for researches working on the interactions between toxin proteins and lipid bilayers: “the structural transition between MBA and porous oligomers is independent from MBA size and could involve formation of a pre-pore”, and “the Q168E mutant is unable to penetrate the membrane and aggregate to form pores”.

I would recommend the paper for publication.

Response. We thank the reviewer for his/her supportive comments and for endorsing our manuscript.

I only have two minor comments:

1. The toxin adsorption kinetics and the final toxin-bilayer structure could depend on the initial toxin concentration. Please, indicate in the text (or in the figure caption) the used toxin concentration.

Response. This information has been added in the Methods section and the sentence now reads “Then, 4 μL of the toxin at a concentration of 4 $\text{mg}\cdot\text{mL}^{-1}$ is injected in the drop.”

2. If possible, show force-distance curves to demonstrate the bilayer existence (which should match with the profile of figure 4i).

Response. Our revised version now includes three force-distance curves (shown in the new Supplementary figure 15) performed on the membrane used in the kinetic experiment on WT Cyt1Aa (before the addition of toxin due to its dramatic effect on membrane stability) as well as in experiments performed with the Q168E mutant and the BSA control (for these, force-distance measurements were feasible even after incubation). All the curves show the puncturing typical of lipid bilayers (indicated with arrows), confirming the stability of the membranes. Perforation occurs over a Z piezo displacement of 7-9 nm which can be translated into a depth of about 5 nm (using a sensitivity value of about 20 nm/V and a spring constant of 0.7 N/m).

Also, other force-distance curves could deliver information about structural changes in the toxin-bilayer with time.

Response. During the kinetics experiment, time was an issue and imaging was not interrupted to perform additional force-distance measurements. We agree with the referee that nanomechanical information could provide further information on the structural changes occurring in the bilayer upon the insertion of toxins. However, at the nanometer scale, it is not

possible to target a specific area on the image to perform the proposed force-indentation measurements. To alleviate this problem, we considered using the quantitative nanomechanical measurement tool of the Multimode 8 device in combination with sharp tips, but we found that such tips cannot be used efficiently on thin membranes. Larger tips would have allowed to accurately measure the proposed mechanics but would have reduced the imaging resolution to the point where frontiers between the pores and the bilayer would have been blurred.

Reviewer #3

This manuscript described the fine-structure of Bti natural parasporal crystal consisting of Cyt1Aa protein at the nano-level by using serial femtosecond nano-crystallography related techniques. Further more, this ms described the structure transition and dynamic of the protoxin, activated toxin and member-bound oligomer, providing many more new and fine knowledge about the structure of parasporal crystal, the processing pathway of protoxin, and the pore-formation.

1. We can know about the native structure of the crystal formed by Bt bacterium. Previously, we just can know the structure of crystal protein in detail. But we can not image the native structure after the crystal protein forms granular crystal. This work showed Cyt1Aa protoxin crystallizes as a domain-swapped dimer with strands $\beta 1$ and $\beta 2$ at the DS interface #1. Even we do not know why this dimer accumulates a bypyramidal shaped crystal.
2. This work described the related concern about the dissolution of crystal and the processing path, which concerning the alpha and beta domain, H-bond, disulfide bridge, N-terminal part as well as pH value and reducing power.
3. This work described how toxin accumulate on the member surface, aggregate as an oligomer and form porous conformer with 54nm diameter by 56 mono proteins.
4. This work use Sf21 insect cells and mammalian cells instead of mosquito midgut cells to elucidate the binding\aggregation\pore formation, and explain why Cyt1A serves as a receptor for 3D domain crystal protein Cry4Ba or Cry11Aa to insertion in membrane (line 406-407).

Response. We thank the referee for his/her positive comments on our work.

If it is right, Cyt1A could also serve as a receptor for 3D domain crystal protein Cry1Aa to insertion in membrane and synergizes Cry1Aa toxin by functioning as a membrane-bound receptor. It is valuable to test this. It is unknown that the Sf cells is same as the mosquito cell when Cyt1A toxin play function on membrane. Even Cyt1A toxin show cytotoxic activity, it only exists in Bti strains which are highly toxic to diptera insects. There is not any Cyt1A toxin existing in Btk strains, which contain 3D domain Cry1A crystal proteins and are highly toxic to lepidoptera insects. If Btk strain obtained cyt1A gene, it would help Btk overcome insect resistance.

Response. To our knowledge, Cyt1Aa is capable of synergizing the activities of Cry11Aa, Cry4Aa and Cry4Ba toxins, with whom it is co-expressed in *Bti* cells, but also that of Cry3Aa from *Bt tenebrionis* (Federici and Bauer 1998, Appl. Env. Microbiol.). Results are more contrasted concerning the ability of Cyt1Aa to synergize Cry1A toxins from *Bt kurstaki*. Synergy of Cyt1Aa with Cry1Ac was evidenced in diamondback moth (Sayyed et al. 2001 Appl. Env. Microbiol.) but it was not found with Cry1Aa in the same species (Meyer et al. 2001 Appl. Env. Microbiol.). Nevertheless, further combinations of Cry1/Cyt surely are worth engineering and testing on different insect pests. The environmentally-safe biopesticidal arsenal represented by *Bt* could be widened by development of new recombinant strains co-expressing toxins of different origin, host spectrum and mode of action. It is our hope that our work, as well as other

studies focusing on the *in cellulo* structures of naturally-crystalline *Bt* toxins, will serve this purpose.

I highly value this work. It will push the work to elucidate the fine structure of all Bt naturally-crystal especially the highly toxic lepidopteran-specific Cry1A toxins, and further to know how and why the crystal protein can naturally form crystalline granule. The crystal protein accounts for 20-30% of the all proteins of dry weight of sporulation culture. If we can let the proteins we need to form crystal inside Bt strain upon the mechanism of crystal formation, we will set up a new expression system as crystal style with Bt system comparing to E.coli system. The crystal structure base on the processing path of crystal in insect gut and the manner of toxin binding, aggerating and perforating will facilitate to know the more detail action model and to help to setup the insect resistance management.

Response. We agree that the establishment of recombinant crystallization in *Bt* would offer a new means to produce recombinant proteins. To our knowledge, this has not been demonstrated yet.

Minor concern:

This work described the natural crystal as a nano-crystal. In my knowledge, nano scale means those less than 100nm. While the size of Bt natural crystal often larger than 100nm and near to 1000 nm. In this work, the Cyt1Aa natural crystal is 925x591 nm as described in table 2.

Response. Crystals are three-dimensional, hence we use the term “nano-crystals” to describe crystals of volume lesser than $1 \mu\text{m}^3$, and Cyt1Aa crystals are $\sim 0.3 \mu\text{m}^3$. Nonetheless we understand why the referee is confused and therefore we have replaced all occurrences of the term “nano-crystal” by “sub-micron-sized crystal”.

REVIEWERS' COMMENTS:

Reviewer #1 (Remarks to the Author):

I am completely satisfied that the authors have all of the key points in the review. I would like to congratulate the authors being able to clearly present such a massive, complex body of work.